# Advances in mixed cell deconvolution enable quantification of cell types in spatial transcriptomic data

Patrick Danaher [1,2 ✉], Youngmi Kim[1,2], Brenn Nelson[1], Maddy Griswold[1], Zhi Yang [1], Erin Piazza [1] & Joseph M. Beechem[1]

Mapping cell types across a tissue is a central concern of spatial biology, but cell type abundance is difficult to extract from spatial gene expression data. We introduce SpatialDecon, an algorithm for quantifying cell populations defined by single cell sequencing within the regions of spatial gene expression studies. SpatialDecon incorporates several advancements in gene expression deconvolution. We propose an algorithm harnessing log-normal regression and modelling background, outperforming classical least-squares methods. We compile cell profile matrices for 75 tissue types. We identify genes whose minimal expression by cancer cells makes them suitable for immune deconvolution in tumors. Using lung tumors, we create a dataset for benchmarking deconvolution methods against marker proteins. SpatialDecon is a simple and flexible tool for mapping cell types in spatial gene expression studies. It obtains cell abundance estimates that are spatially resolved, granular, and paired with highly multiplexed gene expression data.

[1] NanoString Technologies, Seattle, WA, USA. [2]These authors contributed equally: Patrick Danaher, Youngmi Kim. ✉email: pdanaher@nanostring.com

Single-cell RNA sequencing defines the cell populations present within a tissue. But this catalog of cell types begs a question that scRNA-seq cannot answer: how are these cell types arranged within tissues? Spatial gene expression technologies[1,2] measure gene expression within minute regions of a tissue, but do not report an abundance of cell types within these regions, complicating interpretation.

A solution is offered by gene expression deconvolution, a class of algorithms designed to quantify cell populations using gene expression data (Fig. 1). Many algorithms address bulk expression data[3–7], but they are not optimized for the lower signal and higher background of spatial gene expression data. Cell-type quantification in spatial data was first performed using unsupervised factor analysis[8], but this approach eschews pre-specified cell-type expression profiles and therefore loses discriminative power. The first true deconvolution algorithm for spatial data[9] requires scRNA-seq data from matching samples, limiting its use.

Here, we describe SpatialDecon, a toolkit incorporating algorithmic advancements and data resources to make deconvolution of spatial data more accurate and widely applicable. In a benchmarking dataset, we demonstrate superior performance compared to existing methods. In a non-small cell lung tumor, we demonstrate the use of our method to map the cell-type composition and spatial organization of a tumor's immune infiltrate. These measurements reveal the spatial organization of cell types defined by scRNA-seq. Furthermore, they give context to gene-level results, resolving whether a gene's expression pattern reflects differential expression within a cell type or merely differences in cell-type abundance.

## Results

### Log-normal regression improves deconvolution performance.
Gene expression data has extreme skewness and inconsistent variance, but most existing deconvolution algorithms are based in least-squares regression and implicitly assume unskewed data with constant variance[3–5]. We propose to replace the least-squares regression at the heart of classical deconvolution with log-normal regression[10]. This approach retains the mean model of least-squares regression while modeling variability on the log-scale, which largely corrects the skewness and unequal variance of gene expression data in both bulk and spatial experiments

(Supplementary Note and Supplementary Figs. 1 and 2). SpatialDecon, the algorithm implementing this procedure, is described in the Online Methods. Combining a linear-scale mean model and log-scale variability was proposed before as part of the dtangle algorithm[6], which uses only cell-type-specific marker genes. SpatialDecon assumes the same data generating model as dtangle, but its regression framework harnesses all genes, regardless of cell-type specificity.

To compare the performance of log-normal and least-squares deconvolution, two cell lines, HEK293T and CCRF-CEM (Acepix Biosciences, Inc.), were mixed in varying proportions, and aliquoted into a FFPE cell pellet array. Expression of 1414 genes in 700 μm diameter circular regions from the cell pellets were measured with the GeoMx platform.

Four deconvolution methods were run: non-negative least squares (NNLS) and v-support vector regression (v-SVR), which both use a least-squares model; dampened weighted least squares[7] (DWLS), designed for data with unequal variance; and constrained log-normal regression (the log-normal deconvolution algorithm in Online Methods).

Deconvolution accuracy was evaluated by comparing the cell lines' estimated and true mixing proportions (Fig. 2a). The least-squares-based methods NNLS and v-SVR were inaccurate, with respective mean squared error (MSE) of 0.075 and 0.052, and with estimated mixing proportions differing from the truth by as much as 0.41 and 0.47. DWLS and log-normal regression both performed well, with MSE's of 0.038 and 0.009, and with maximum errors of 0.16 and 0.12.

**Least-squares deconvolution is statistically inefficient.** To investigate the poor performance of least-squares-based methods, we measured the influence of each gene on deconvolution results from a single region with an equal mix of HEK293T and CCRF-CEM. Each gene's influence was measured as the difference in estimated HEK293T proportion using the complete gene set compared to a leave-one-out set omitting the gene in question.

The least-squares methods NNLS and v-SVR both had genes with high influence on deconvolution results, while DWLS and the log-normal method were not subject to outsize influence from any genes (Fig. 2b). For NNLS, a single high-expression gene changed the model's estimated mixing proportion from 66 to 22%, a

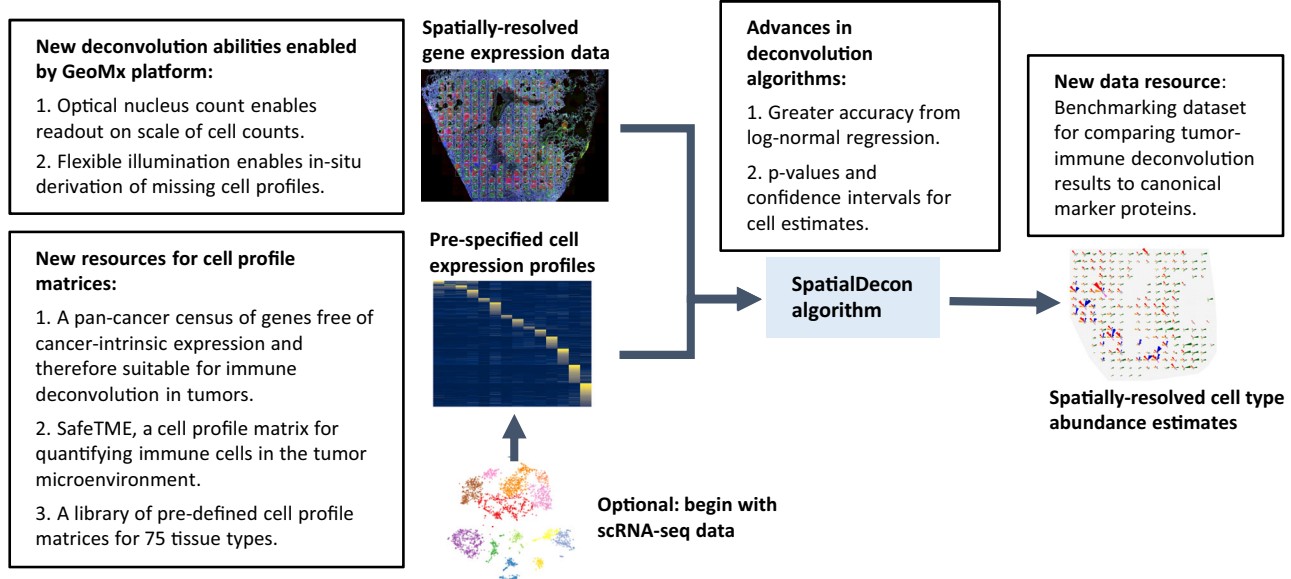

**Fig. 1 Overview of algorithm and advancements to the deconvolution field.** The image summarizes the deconvolution workflow. Text boxes summarize developments proposed in this manuscript. Source data are provided as a Source Data file.

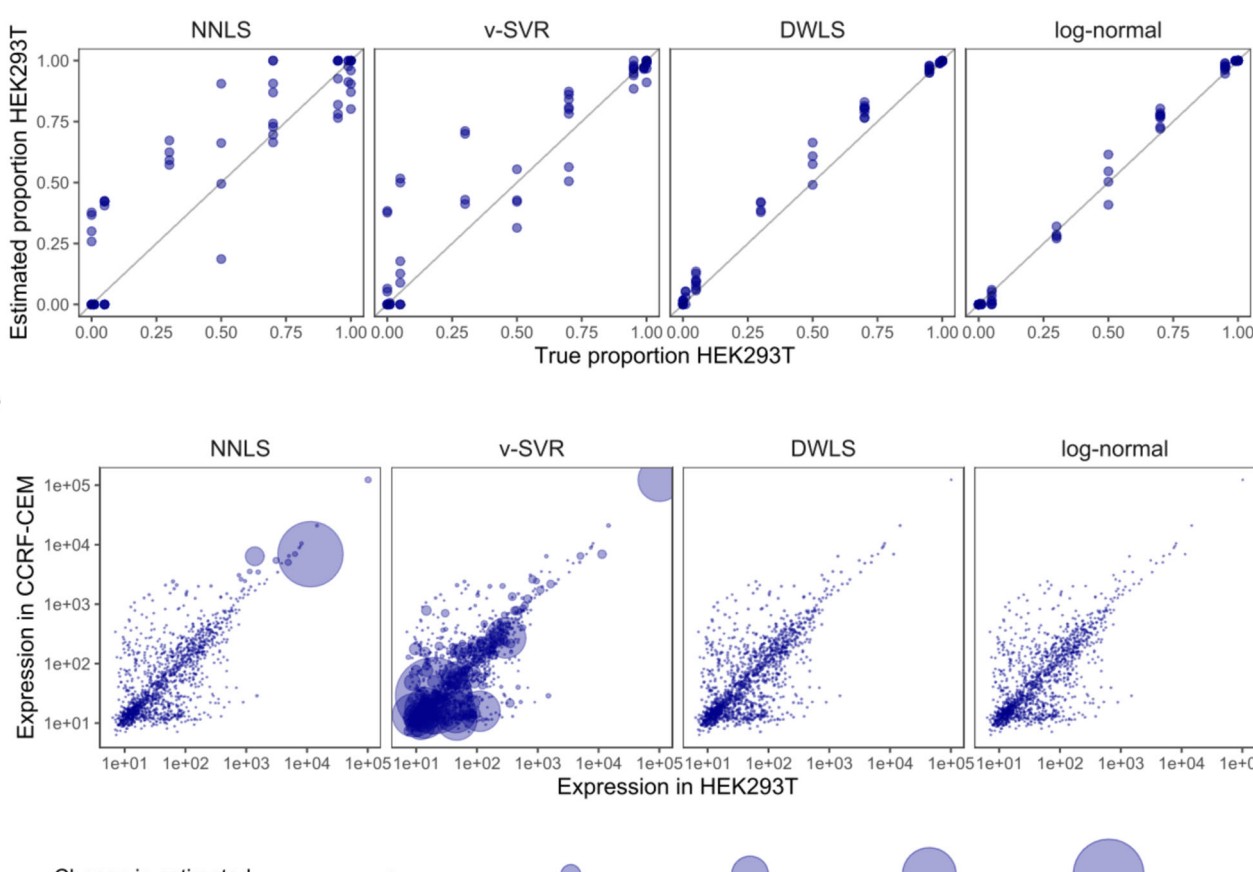

**Fig. 2 Comparison of deconvolution algorithms in mixtures of two cell lines.** The cell lines HEK293T and CCRF-CEM were mixed in varying proportions and profiled with the GeoMx platform. **a** True mixing proportions plotted against estimates from four deconvolution algorithms: non-negative least squares (NNLS), v-support vector regression (v-SVR), Dampened Weighted Least Squares regression (DWLS), and log-normal regression. **b** Influence of each gene on the deconvolution result from a single cell pellet with a 50–50 mix. Point size shows the change in estimated mixing proportion when each gene is removed. Source data are provided as a Source Data file.

remarkable impact on a fit derived from 1414 genes. In v-SVR, high-influence genes were found across all expression levels; the most influential gene changed the estimated proportion from 31 to 83%. In contrast, removing the highest-influence gene from the log-normal and DWLS models changed the estimate by less than 0.3%.

**Pan-cancer screen for genes suitable for tumor-immune deconvolution.** Deconvolution of immune cells in tumors encounters another complication: genes expressed by cancer cells contaminate the data, causing overestimation of the immune populations also expressing those genes. We analyzed 10,377 TCGA samples to identify a list of genes with minimal contaminating expression by cancer cells. We used marker genes[11,12] (Supplementary Table 1) to score abundance of immune and stromal cell populations in each sample, and we modeled each gene as a function of these cell scores. For each gene, these models estimated the proportion of transcripts derived from cancer cells compared to immune and stromal cells in the average tumor (Supplementary Table 2).

Genes exhibited a wide range of cancer-derived expression (Fig. 3a). Across all non-immune cancers, 5844 genes had less than 20% of transcripts attributed to cancer cells. Confirming the stability of this analysis, estimates of cancer-derived expression were largely consistent across TCGA datasets (Fig. 3b).

Confirming the specificity of this analysis, canonical marker genes were consistently estimated to have low percentages of transcripts from cancer cells. Gene lists used in many popular immune deconvolution algorithms[3,12–16], most of which were designed for use in PBMCs and not in tumors, include numerous cancer-expressed genes (Fig. 3c).

**SafeTME: a cell profile matrix for tumor-immune deconvolution.** To support deconvolution of the tumor microenvironment, we assembled the SafeTME matrix, a cell profile matrix for the immune and stromal cell types found in tumors. This matrix combines cell profiles derived from flow-sorted PBMCs[5], scRNA-seq of tumors[17], and RNA-seq of flow-sorted stromal cells[18]. It includes only genes estimated by the above pan-cancer analysis to have less than 20% of transcripts attributed to cancer cells.

**A library of cell profile matrices for diverse tissue types.** To facilitate cell-type deconvolution in diverse tissue types, we derived cell profile matrices from 75 publicly available scRNA-seq datasets[19–37] (Supplemental Table 3). These profile matrices include 17 adult human tissues, 15 human fetal tissues, 6 SARS-CoV-2-infected human tissues, 24 adult mouse tissues, 6 neonatal mouse tissues, and 7 fetal mouse tissues. From each dataset, we

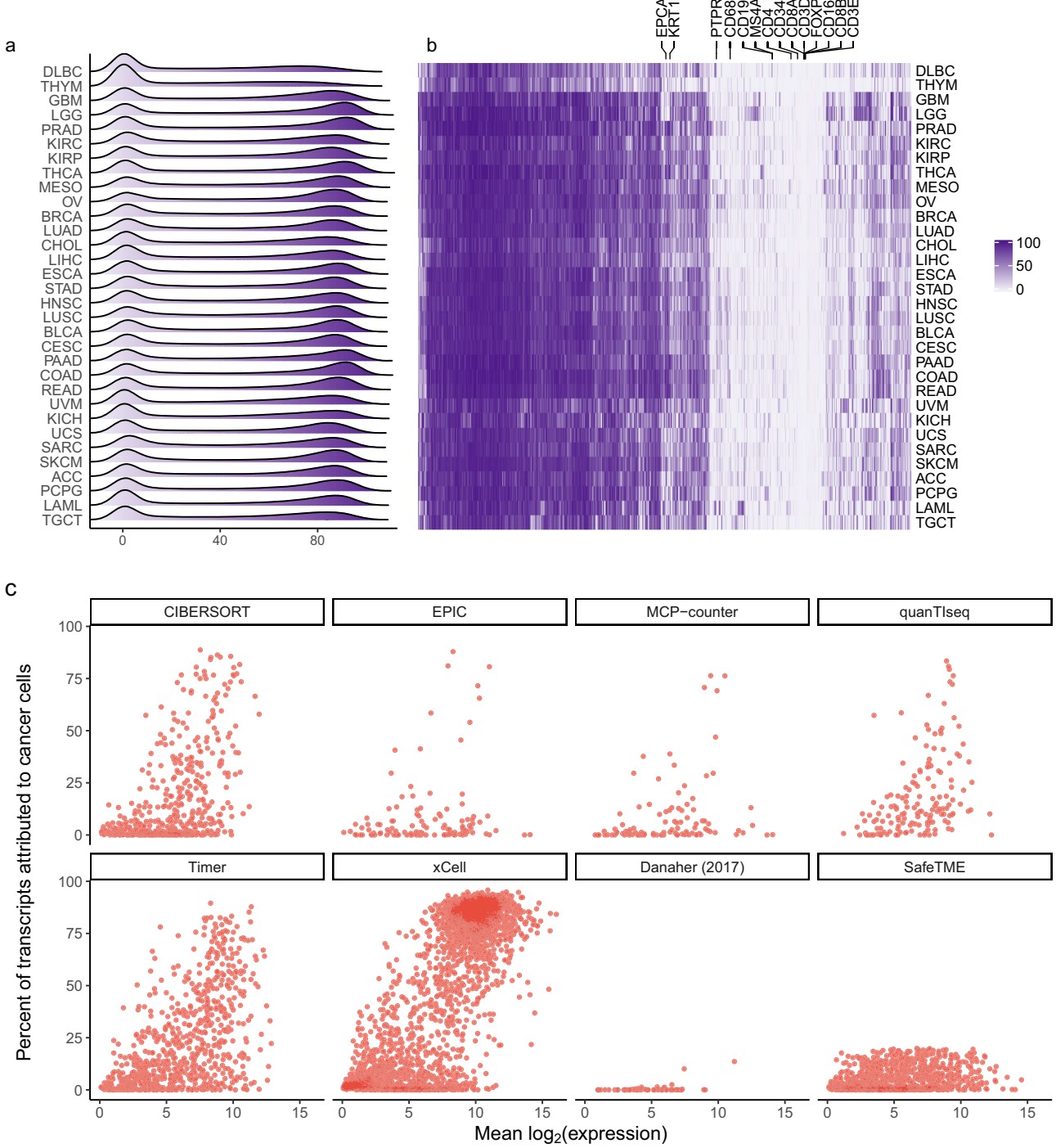

**Fig. 3 Genes' proportions of cancer cell-derived expression in tumors. a** For each cancer type, density of genes' percent of transcripts attributed to cancer cells. **b** For all genes in all cancer types, estimated percent of transcripts attributed to cancer cells. **c** Averaged across all non-immune tumors, genes' mean expression and percent of transcripts attributed to cancer cells. Panels show gene lists from CIBERSORT, EPIC, MCP-counter, quanTIseq, Timer, xCELL, Danaher (2017), and SafeTME, the tumor-immune deconvolution cell profile matrix developed here. Source data are provided as a Source Data file.

compiled the mean expression profile of each cell type, using the cell-type classifications of the original paper.

**Harnessing the GeoMx platform to enhance deconvolution.** The GeoMx DSP platform extracts gene or protein expression readouts from precisely targeted regions of tissue. First, the tissue is stained with up to four visualization markers, and a high-resolution image of the tissue is captured. Using this image, precisely defined

segments of the tissue can be selected for expression profiling; regions can be as small as a single cell or as large as a $700 \times 800$ μm region, and they can have arbitrarily complex boundaries. This flexibility in defining areas to be sampled is often used to split regions of tissue into two segments, e.g., a PanCK+ cancer cell segment and a PanCK- microenvironment segment.

Two features of the GeoMx platform expand the abilities of mixed cell deconvolution. First, the GeoMx platform can profile

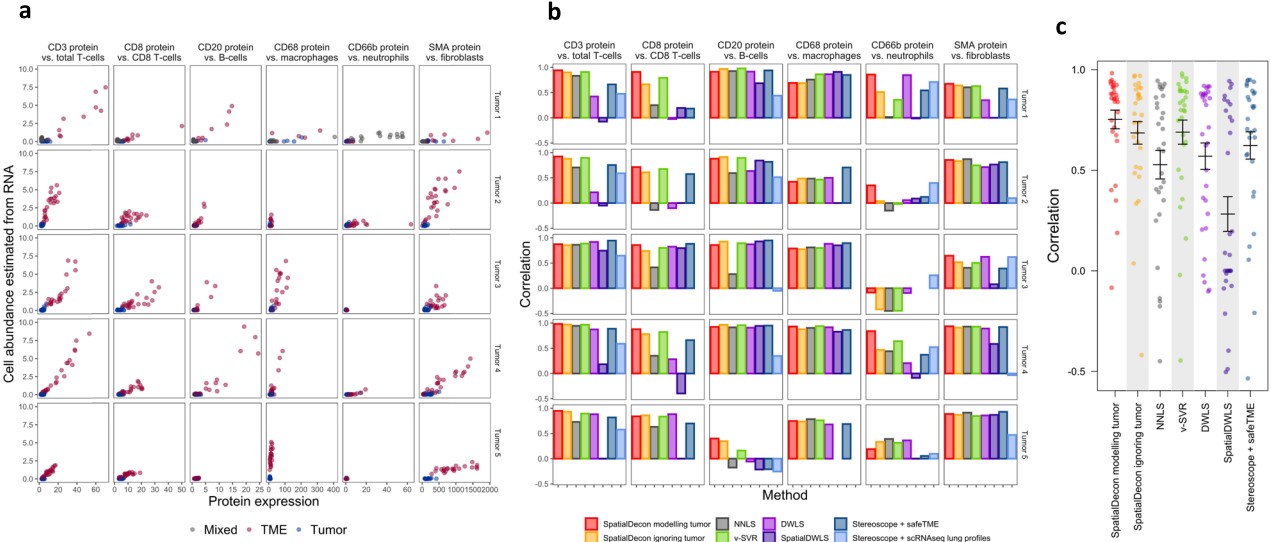

**Fig. 4 Benchmarking of immune deconvolution against the expression of canonical marker proteins. a** Expression of marker proteins (horizontal axis) against cell abundance estimates from the application of SpatialDecon to gene expression data (vertical axis). Each column of panels shows results from a single protein/cell pair; each row shows results from a different lung tumor. Tumor segments are shown in blue, microenvironment segments in red. **b** Pearson correlation between protein and cell abundance estimates for different deconvolution algorithms. NNLS non-negative least squares, v-SVR v-support vector regression, DWLS Dampened Weighted Least Squares. CD8 T cells and macrophages are unavailable under the Stereoscope + scRNA-seq lung profiles method. **c** Mean correlations between deconvolution methods and protein expression. Lines show 95% confidence intervals from $n = 30$ correlations. Source data are provided as a Source Data file.

and model cell types that are absent in the pre-defined cell profile matrix. For example, when performing immune deconvolution in tumors, the expression profile of the cancer cells is often unknown. In such cases, the GeoMx platform can be used to select and profile regions of pure cancer cells, and this study-specific cancer cell profile can be merged with the pre-defined cell profile matrix. This method is used to account for cancer cell expression in the deconvolution analyses of Figs. 4 and 5.

Second, the platform counts the nuclei in every tissue segment it profiles. This nuclei count lets SpatialDecon estimate not just proportions but absolute counts of cell populations. The results of Fig. 5 show cell population count estimates derived in this manner.

**Benchmarking with paired spatial RNA and protein expression**. Due to practical limitations, most experiments benchmarking the performance of immune cell deconvolution methods rely on simulated data, generated either by in silico mixing of cell expression profiles[38] or by in vitro mixing of purified cell populations[39]. However, simulations cannot faithfully represent performance in tumor samples: immune cell expression differs between blood and tumors[17], and cancer cells can express putative immune genes. To benchmark deconvolution performance in real tumor samples, we used the GeoMx platform to collect paired measurements of gene expression and of canonical marker proteins. To our knowledge, the resulting dataset assesses performance over more cell types than any other non-simulated benchmarking dataset.

From five FFPE lung tumors, we took two adjacent slides. We selected 48 700 -μm regions from the first slide, and we identified their corresponding regions in the second slide. The selected regions in the first slide were profiled with the GeoMx protein assay, and the corresponding regions in the second slide were profiled with the GeoMx RNA assay, measuring 1700 genes, including 544 from the SafeTME matrix. Within each region, the GeoMx system's flexible segmentation capabilities were used to collect separate profiles for tumor cells and for microenvironment cells.

**SpatialDecon outperforms alternative methods**. SpatialDecon was run on the benchmarking RNA data using the SafeTME matrix and the unknown cell-types algorithm (Online Methods) to model tumor expression. We compared its cell abundance estimates with the expression of canonical marker proteins (Fig. 4a and Supplementary Table 4). In the average tissue, the Pearson correlation between protein expression and estimated cell abundance was 0.93 for CD3 protein and T cells; 0.84 for CD8 protein and CD8 T cells; 0.72 for CD68 protein and macrophages; 0.80 for CD20 protein and B cells; and 0.80 for SMA protein and fibroblasts. Neutrophils, whose low abundance in many tissues limited the range over which correlation could be observed, achieved an average correlation of just 0.43 with CD66b protein. However, in the two samples with the highest estimated neutrophils, this correlation rose to 0.86 (Tumor 1) and 0.84 (Tumor 4).

To benchmark SpatialDecon, we ran alternative methods: NNLS, v-SVR, DWLS[7], SpatialDWLS[40], and Stereoscope[9]. SpatialDecon was run a second time without using the unknown cell-types algorithm to model tumor expression. All methods used the SafeTME profile matrix. To mimic Stereoscope's recommended approach for tumors without scRNA-seq data, Stereoscope was run a second time using cell profiles derived from a lung scRNA-seq study.

For each cell type and each tissue, we recorded the Pearson correlation between deconvolution results and protein expression (Fig. 4b, c). SpatialDecon performed best, with an average correlation exceeding all other methods by at least 0.06 (compared to v-SVR, paired $t$ test $P = 0.024$, 95% confidence interval = (0.009, 0.117), df = 29) and by as much as 0.22 (compared to NNLS, $P = 0.00018$, 95% confidence interval = (0.118, 0.332), df = 29). Stereoscope had higher correlations using the SafeTME matrix that it did using parameters derived from lung scRNA-seq ($P = 0.038$, 95% confidence interval = (0.014, 0.419), df = 19). SpatialDecon performed better when using the unknown cell-types algorithm to model tumor expression ($P = 0.009$, 95% confidence interval = (0.018, 0.116),

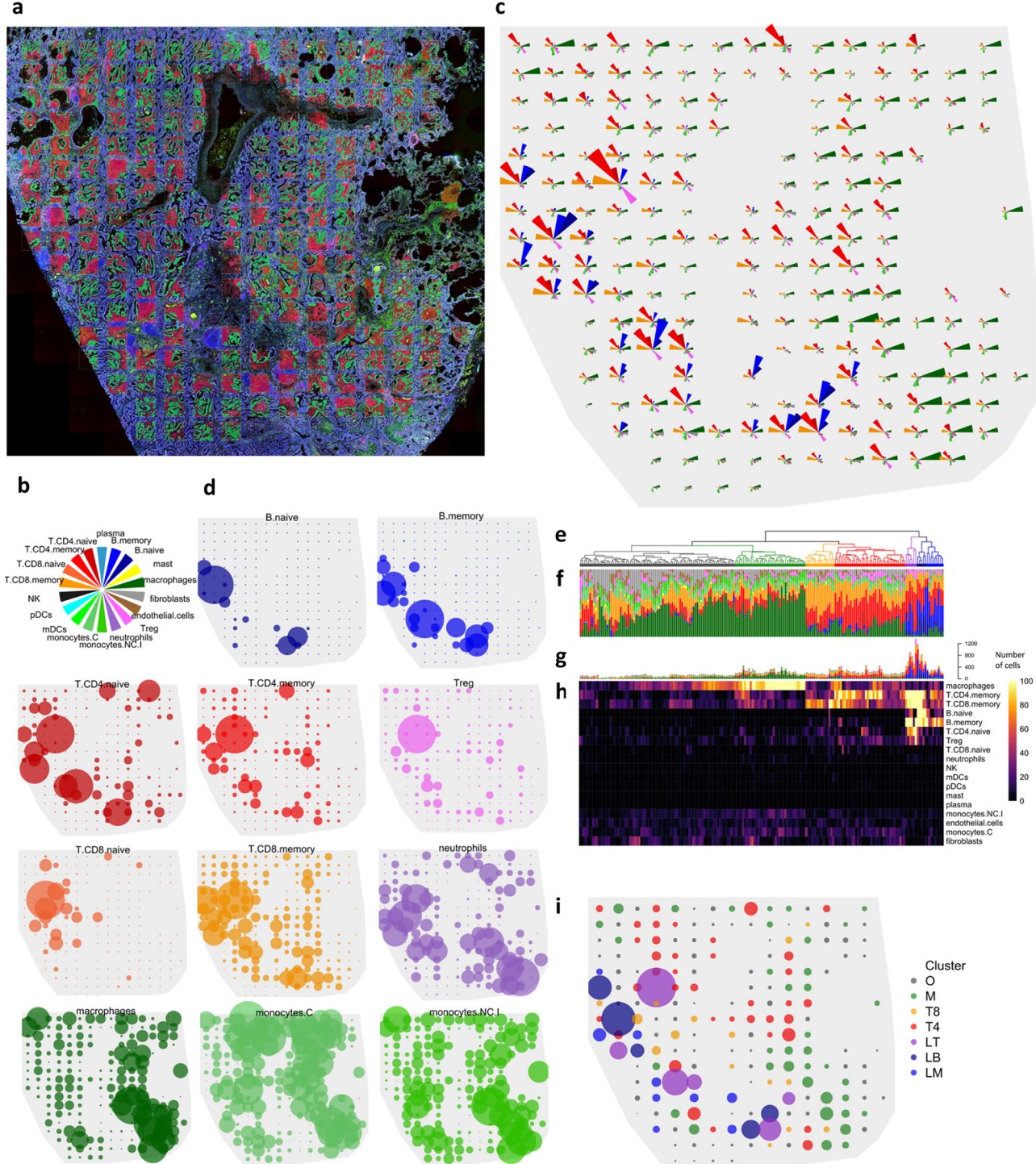

**Fig. 5 Immune cell deconvolution in 191 microenvironment segments of a NSCLC tumor. a** Image of the tumor, with segments superimposed. Green = Pan-cytokeratin+ (tumor) segments; red = Pan-cytokeratin− (microenvironment) segments. **b** Color key for panels **c**, **d**, **f**, and **g**. **c** Abundance estimates of 18 cell types in the microenvironment segments within 191 regions of the tumor. Wedge size is proportional to estimated cell counts. **d** Abundance estimates of 12 cell populations in microenvironment segments. Point size is proportional to estimated cell counts within each panel; scale of point size is not consistent across panels. **e** Dendrogram showing clustering of microenvironment segments' abundance estimates. **f** Proportions of cell populations in microenvironment segments. **g**, **h** Estimated absolute numbers of cell populations in microenvironment segments. **i** Spatial distribution of microenvironment segment clusters. Point color indicates cluster from (**e**); point size is proportional to total estimated immune and stromal cells in microenvironment segments. Source data are provided as a Source Data file.

df = 29). SpatialDWLS performed well in some comparisons but failed in others.

**Mapping the immune infiltrate in a non-small cell lung cancer.** As a demonstration of spatial gene expression deconvolution, immune cell abundances were estimated across a grid of 191 regions of a non-small cell lung cancer (NSCLC) tumor. The GeoMx RNA assay was used to measure 1700 genes, including 544 genes from the SafeTME matrix. The tissue was stained with fluorescent markers for PanCK (tumor and epithelial cells), CD45 (immune cells), CD3e (T cells) and DNA. In total, 191 300 × 300 μm regions of interest were arrayed in a grid across the 7.8 × 6.7 mm span of the tumor. Within each region of interest, the flexible illumination capability of the GeoMx platform was used to separately assay two segments: tumor segments, defined by PanCK+ stain, and microenvironment segments, defined as the tumor segments' complement (Fig. 5a).

The SpatialDecon algorithm was applied to all segments in the dataset using the SafeTME matrix along with tumor-specific profiles derived from the study's PanCK+ segments. On a 1.9 Ghz laptop, deconvoluting 376 segments took 29 s. Using just the microenvironment segments, we assembled a map of the tumor's immune infiltrate (Fig. 5c, d). The most abundant cell types were CD4 T cells (11,943 across all segments), macrophages (10,055), and CD8 T cells (7405). The algorithm estimated very low immune cell content in the tumor segments, with a mean of 7 immune and stromal cells per tumor segment, compared to a mean of 216 immune and stromal cells per microenvironment segment.

Cell populations had distinct spatial distributions. Naïve and Memory B cells had the most concentrated spatial distributions (Gini coefficients = 0.84, 0.85), localizing primarily within a band of regions on the left side of the tumor. Naive, memory and regulatory CD4 T-cell populations (Gini = 0.70, 0.64, and 0.57) had many dense foci near the B-cell-enriched regions and sporadic foci elsewhere in the tumor. Naive CD8 T cells (Gini = 0.52) were concentrated in the top-right of the tumor, while memory CD8 cells were present throughout the tumor. Macrophages (Gini = 0.48) and non-conventional/intermediate monocytes (Gini = 0.45, 0.39) were enriched in the lower-right of the tumor, away from the B cells and T cells, while conventional monocytes (Gini = 0.41) were enriched in the upper-right. Neutrophil-enriched segments (Gini = 0.41) appeared in both lymphoid-rich and myeloid-rich areas.

Hierarchical clustering on cell abundances identified seven subtypes of tumor microenvironment regions (Fig. 5e). The largest cluster, Subtype O, was defined by low total numbers of immune cells and consisted primarily of macrophages, memory CD8 T cells, monocytes, and fibroblasts. Subtype M was dominated by macrophages. Subtype T8 was dominated by memory CD8 T cells, with less abundant memory CD4 T cells. Subtype T4 was dominated by memory CD4 T cells, with less abundant memory CD8 T cells. Subtype LT consistent almost entirely of lymphoid cells, with majority T cells but also abundant memory B cells. Subtype LB also consistent almost entirely of lymphoid cells but had higher proportions of B cells, both memory and naive. Subtype LM was lymphoid-dominated but had as much as 15% macrophages. Each subtype was concentrated within, but not confined to, a distinct area of the tumor.

**Reverse deconvolution gives context to gene expression results.** Variability in gene expression is driven both by changing abundance of cell populations and by differential regulation within cells. These two sources of variability can be decomposed via reverse deconvolution, in which each gene's expression is predicted from cell abundance estimates. Outputs of this reverse deconvolution include genes' fitted expression values based on cell abundances, and their residuals, calculated as the log2 ratio between observed and fitted expression (Fig. 6a). These residuals measure genes' up- or downregulation within cells, independent of cell abundance.

To interpret our gene expression data in the face of highly variable cell mixing, we fit reverse deconvolution models over the microenvironment segments of the NSCLC tumor from Fig. 5. Each gene's dependency on cell mixing was measured with two metrics: the Pearson correlation between observed and fitted expression, and the standard deviation of the residuals. Based on these metrics, genes fell into four categories, each with a different implication for analysis and interpretation of genes' data (Fig. 6b). Genes with low correlations and high residual standard deviations, e.g., *MT1M*, are mostly independent of cell-type mixing and can be understood without reference to cell abundances (Fig. 6c). Genes with low correlations and low residual standard deviations, e.g., *ARG1*, have little variability to analyze. Genes with high correlations and low residual standard deviations, e.g., *PDCD1*, merely provide an obtuse readout of cell-type abundance. Genes with high correlations and high residual standard deviations, e.g., *CCL19*, have substantial variability unexplained by cell mixing, but this variability is concealed by even greater variability driven by cell mixing. Analysis of these genes' residuals reveals the full complexity of their behavior. For example, *CXCL13* expression was over twofold higher or lower than expected in some regions (Fig. 6d). *LYZ* expression, 84% of which was attributed to macrophages and monocytes, was highest in a corner of the tumor where those cell populations had relatively low abundance (Fig. 6e). *CCL17* was highly expressed in sporadic regions across the tumor, and in most of these regions the high expression was beyond what cell abundance alone could explain (Fig. 6f).

**Residuals of reverse deconvolution reveal co-expressed genes.** Cell mixing induces correlation between genes that are expressed by the same cell type but that are not otherwise co-expressed at the cellular level. In the residuals of reverse deconvolution, this unwanted correlation abates, revealing correlation induced by coordinated expression within cell types (Fig. 6g). For example, the Pearson correlation between *CD8A* and *CD8B* was 0.75 in the log2-scale data from microenvironment segments; in residual space, their correlation was −0.03. Pearson correlation between *MS4A1* and *CD19* was 0.82 in the normalized data and 0.06 in residual space.

To identify candidate co-expressed genes, we identified gene clusters with high Pearson correlation in residual space. A cluster of six HLA genes varied smoothly across the tissue, weakly correlated with macrophage abundance but also elevated in many macrophage-poor regions (Fig. 6h). In the two regions with the most macrophages, these genes all had negative residuals, suggesting suppressed antigen presentation by macrophages in those regions. It has been previously shown that these HLA genes share regulatory elements[41]. Another cluster consisted of lipid metabolism and small molecule transport genes (*ACP5*, *APOC1*, *ATP6V0D2*, *CYP27A1*, *LIPA*). Absolute expression of these genes was elevated in the tissue's lower-right corner. Analysis of residuals reveals additional spatial expression dynamics, including a region of up-regulation in the upper-left side of the tissue and a region of downregulation in the lower-left (Fig. 6i).

**Deconvolution of granularly defined cell types.** It is commonplace for scRNA-seq studies to report multiple sub-clusters for a given cell type. To assess SpatialDecon's ability to deconvolve

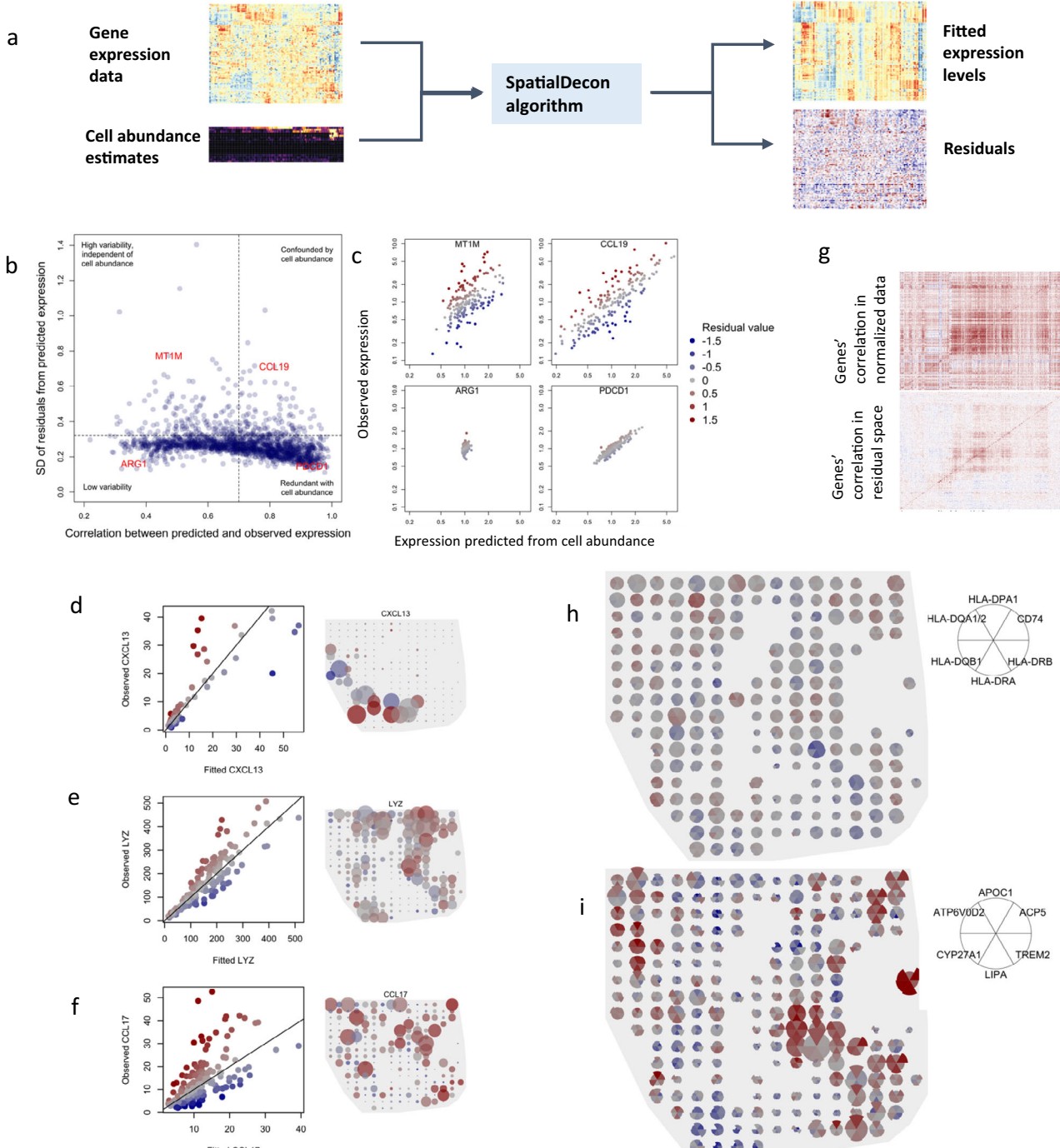

**Fig. 6 Results of reverse deconvolution in a NSLCL tumor. a** Schematic of reverse deconvolution approach: gene expression is predicted from cell abundance estimates using the SpatialDecon algorithm, obtained fitted values and residuals. **b** Genes' dependence on cell mixing. The horizontal axis shows Pearson correlation between observed expression and fitted expression based on cell abundance. Vertical axis shows the standard deviation of the log2-scale residuals from the reverse deconvolution fit. **c** Example genes from the extremes of the space of panel (**b**) are shown, with observed expression (vertical axis) plotted against fitted expression (horizontal axis). Color scale applies to panels **c, d, e, f, h, i**. **d–f** For CXCL13, LYZ and CCL17, observed expression is plotted against fitted expression (left), and observed expression is plotted in the space of the tissue (right). In all panels, point color indicates residuals. In panels on the right, point size is proportional to observed expression level. **g** Pearson correlation matrices of genes in log-scale normalized data (top) and in residual space (below). **h, i** Spatial expression of gene clusters defined by high correlation in residuals of reverse deconvolution. Wedge color shows genes' residual values; wedge size is proportional to genes' expression levels. Source data are provided as a Source Data file.

such closely-related cell types, we created a second, more granular cell profile matrix. Using the same genes as the SafeTME, we merged profiles from 42 immune cell sub-clusters defined in a scRNA-seq study[17] with the flow-sorted fibroblast and endothelial cell profiles[18].

To confirm the stability of SpatialDecon when using granular cell profile matrices, we re-ran the benchmarking analysis of Fig. 4a with this expanded matrix. We then compared marker protein expression with the total abundance of relevant cell types, for example comparing CD68 protein to the total abundance

scores of the matrix's nine macrophage sub-cluster profiles. Concordance between deconvolution and marker proteins was similar to the results achieved by the SafeTME matrix (Supplementary Fig. 6).

To show the utility of SpatialDecon with granular cell profile matrices, we used the expanded profile matrix to re-run deconvolution of the NSCLC shown in Figs. 5 and 6 (Supplementary Fig. 7). The abundance of some cell types was dominated by a few sub-clusters; for example, 83% of total macrophages belonged to clusters hMac5 and hMac7. In contrast, T-cell sub-clusters were more evenly represented, with each of the seven sub-clusters representing between 5 and 26% of total T cells. Sub-clusters' spatial distributions were distinct and consistent with known biology. For example, the cycling T-cell cluster, hT7, was primarily present in the B-cell enriched regions. The dense T cells and B cells of these regions suggest they are tertiary lymphoid structures, where T cells are activated and prompted to proliferate[42]. Another T-cell cluster confined to these regions was hT4, which is defined by the expression of CXCL13, a B-cell chemoattractant pivotal in forming and maintaining tertiary lymphoid structures[43]. Finally, the cytotoxic CD8 T-cell cluster, hT1, and the Treg cluster, hT3, invaded the same regions of the tissue, consistent with their demonstrated tendency to traffic together[44].

## Discussion

Cell deconvolution promises to be a linchpin of spatial gene expression analysis. Cell abundance estimates offer a functional significance and ease of interpretation unmatched by gene expression values. Cell abundance also gives context to gene expression results, disambiguating whether a gene's expression pattern results from differential cell-type abundance or differential expression within cell types.

The methods described here enable spatial studies as a natural follow-on to scRNA-seq: given cell populations defined by scRNA-seq, deconvolution in spatial gene expression data reveals how those cells are arranged within tissues, obtaining a region-by-region accounting of their abundance. This allows new questions to be asked: How are cell types arranged and mixed with each other? Which cell types repel or attract each other? Which cell types explain the expression pattern of a gene of interest? How does a cell population's behavior change when it is co-localized with another cell population?

While developing SpatialDecon, we found that modeling background counts were critical to the accurate deconvolution of spatial data. In bulk mRNA studies, where signal-to-background ratios are stronger, we anticipate that modeling of background will be helpful but often not essential.

Historically, gene expression deconvolution methods required a cell profile matrix, usually derived from flow cytometry or scRNA-seq. Many recent methods[45,46] instead take a scRNA-seq dataset as input, effectively automating the derivation of cell profile matrices. However, requiring scRNA-seq data limits these methods' use to experiments where such data is available. This limitation is most acute in cancer studies, where the heterogeneity of tumor gene expression means public scRNA-seq data will not be representative of a new tumor's spatial data. In contrast, SpatialDecon's use of a cell profile matrix makes it applicable across the larger set of studies where either scRNA-seq data or a cell profile matrix is available. Critically, this includes any solid tumor study where immune cells are of interest.

The methods and data resources described here promise to improve deconvolution not just in spatial expression data but also in bulk gene expression. Log-normal regression has the same theoretical benefits in bulk expression deconvolution. Our library of cell profile matrices for diverse tissues directly supports deconvolution in bulk gene expression experiments. Finally, future attempts to deconvolve immune cells in bulk tumor expression data should confine the analysis to our list of genes not expressed by cancer cells.

Based on cell abundances, we identified seven microenvironment subtypes within one NSCLC tumor. This heterogeneity raises the prospect that tumors could be classified not just by their overall cell abundance, but by the localized microenvironment subtypes they contain.

## Methods

**Acquisition of human samples**. This research complied with all relevant ethical regulations. All human tissue samples were sourced from ProteoGenex Inc, a commercial tissue vendor. Informed consent was provided for all samples, and ProteoGenex asserts that all specimens were collected under ethical regulations and in accordance all applicable (local and international) laws. Written consent was given for the broad research use of these samples and not specifically for this study. Clinical information for these samples is in Supplementary Table 5.

**The SpatialDecon algorithm**:
Notation:
Use "observation" to denote a tissue region from which a gene expression profile is collected.
Let $i$ index observations, let $j$ index genes, and let $k$ index cell types.
Let $X_{p \cdot K}$ be the cell profile matrix giving the linear-scale expression of $p$ genes over $K$ cell types.
Let $Y_{p \cdot n}$ be the observed expression matrix of $p$ genes over $n$ observations.
Let $\beta_{K \cdot n}$ be the unobserved matrix of cell type abundances of $K$ cell types over $n$ observations.
Let $B_{p \cdot n}$ be the matrix of expected background counts corresponding to the elements of $Y$.
Let $||\mathbf{x}||$ denote the operator of a vector $\mathbf{x}$ such that $||\mathbf{x}|| = \text{mean}(\mathbf{x}^2)$.
The core log-normal deconvolution algorithm proceeds as described in Box 1.
To guard against errors in the cell profile matrix and noise in the data, the SpatialDecon algorithm incorporates outlier removal into the log-normal deconvolution algorithm. Outlier removal improves deconvolution accuracy in simulated data (Supplementary Figs. 9 and 10). The SpatialDecon algorithm proceeds as described in Box 2.

**Estimating background**. For GeoMx studies, each region's background level can be estimated by taking the mean of the negative control probes. These probes target sequences identified by the External RNA Controls Consortium (ERCC) as alien to the human genome.

---

**Box 1 | Log-normal deconvolution algorithm**

1. To avoid negative-infinity values when log-transforming zero-valued elements of $Y$, define $\varepsilon$ equal to the minimum non-zero value in $Y$, and threshold $Y$ below so that its smallest value is $\varepsilon$. Supplementary Fig. 8 demonstrates the algorithm's robustness to the choice of $\varepsilon$.
2. For $i$ in $\{1, ..., n\}$, take $\widehat{\boldsymbol{\beta}}_{\cdot i} = \text{argmin}_{\beta_{\cdot i}} ||\log(\mathbf{Y}_{\cdot i}) - \log(B_{\cdot i} + X\beta_{\cdot i})||$, subject to the constraint that $\boldsymbol{\beta}_{\cdot i} \geq 0$. This constrained optimization is performed separately for each column of $Y$ using the R package logNormReg[9]. This step assumes that different columns of $Y$, i.e., different tissue regions, are statistically independent. To the extent that this assumption fails, greater accuracy could be gained by modelling the dependence between regions. Modelling this dependence structure is a problem left for future algorithms; to our knowledge, no attempts to do so have been described.
3. For $i$ in $\{1, ..., n\}$, calculate the covariance matrix of $\widehat{\boldsymbol{\beta}}_{\cdot i}$ by inverting the Hessian matrix returned by logNormReg. Call this covariance matrix $\widehat{\sum}^{(i)}$.
4. Return $\widehat{\beta}$ and $\widehat{\sum}^{(i)}$.

---

---

**Box 2 ▌ The SpatialDecon algorithm**

1. Run the log-normal deconvolution algorithm.
2. Choose $\delta$ as the expression level below which technical noise predominates. For GeoMx data normalized to have expected background $= 1$, we use $\delta = 0.5$.
3. Define the residuals of the algorithm fit as $R = \log_2(\mathrm{pmax}(Y, \delta)) - \log_2(\mathrm{pmax}(B + X\hat{\beta}, \delta))$, where $\mathrm{pmax}(x, \delta)$ is the function replacing all elements of $x$ below $\delta$ with $\delta$.
4. For all $\{i, j\}$ with $|R_{i,\,j}| > 3$, set $Y_{i,j}$ to NA. In simulated data, this threshold of 3 performed well, and deconvolution results were not sensitive to the choice of threshold (Supplementary Fig. 9).
5. Re-run the log-normal deconvolution algorithm using the updated $Y$ matrix, obtaining estimates $\hat{\beta}$ and covariance matrices $\widehat{\textstyle\sum}^{(i)}$.
6. Calculate the standard error for each $\hat{\beta}_{j,i}$ with $\mathrm{sqrt}\,(\widehat{\textstyle\sum}_{j,j}^{(i)})$.
7. Calculate the p-value for each $\beta_{i,j}$ with $p = 2\,(1 - \phi(t = \hat{\beta}_{j,i},\ df = p - K - 1))$, where $\phi$ is the cumulative distribution function of the standard normal distribution. ($\hat{\beta}_{j,i}/\widehat{\textstyle\sum}_{j,j}^{(i)}$ is the square root of the Wald statistic, which is asymptotically normal[51].)
8. Calculate the 95% confidence interval for each $\beta_{i,j}$ with $\hat{\beta}_{j,i} \pm 1.96\,(\widehat{\textstyle\sum}_{j,j}^{(i)})^{1/2}$

Return $\hat{\beta}$ along with the standard errors and p-values of its elements.

---

**Box 3 ▌ Unknown cell-types algorithm**

1. Specify columns of $Y$ corresponding to segments selected to contain a pure cell type that is missing from $X$. For example, for immune deconvolution in tumors, select segments targeting purely PanCK+ cells to derive a cancer cell profile.
2. Collapse the segments into $G$ clusters by applying the R functions hclust and cutree to their log-transformed expression profiles. The value of $G$ should be set large enough to capture the diversity of the segments of the unmodelled cell type and small enough to avoid adding excessive terms to the deconvolution model. For healthy cell types with homogeneous behavior, 1–2 clusters is adequate; for tumor cells, which can be highly heterogeneous, a higher $G$ is recommended. We use $G = 10$ in the tumor analyses in this manuscript. Supplementary Fig. 11 shows the choice of G to have a minimal impact on performance.
3. Define each cluster's expression profile by taking each gene's geometric mean across the observations in the cluster. To avoid convergence failures, scale this profile similarly to the other columns of the cell profile matrix such that its 90th percentile is equal to the average 90th percentile of the other columns.

Append the cluster expression profiles to the cell profile matrix.

---

**Application to Visium data.** For Visium studies, background should be set slightly above zero to account for the diffusion of barcodes across tiles; a value of 0.01 would correspond to a belief in negligible diffusion, while a value of 1 would guard against substantial diffusion. In an analysis of a Spatial Transcriptomics dataset from a breast cancer[47], the choices of 0.01 and 1 produced highly similar deconvolution results, with Pearson correlation between abundance scores equal to 0.97, and with results matching expected biology (Supplementary Fig. 12). In a Visium dataset from an ovarian cancer, setting background at 0.01 produced results consistent with known biology (Supplementary Fig. 13). Alternatively, more accurate deconvolution could be achieved using a background matrix based on each spot's estimated burden of counts diffusing from neighboring spots.

**Using the GeoMx platform to derive cell profiles for unmodelled cell types.** Many tissues will have an unmodelled cell type—a cell type known to be present in the tissue but that is missing from the cell profile matrix. The presence of tumor cells when performing immune deconvolution is the most common instance of unmodelled cells.

The procedure for using GeoMx to derive the profiles of unmodelled cells and merge them into the cell profile matrix $X$ proceeds as described in Box 3.

The scaling operation in step 3 is arbitrary. Therefore, it is not recommended to directly compare tumor abundance scores derived with this method to immune abundance scores. This algorithm should primarily be considered a tool for obtaining more accurate immune abundance scores.

The NanoString GeoMx® Digital Spatial Profiler and GeoMx assays are for research use only and not for use in diagnostic procedures.

**Converting abundance scores to cell counts.** When the GeoMx system's per-region nuclei counts are available, the below procedure converts cell abundance scores to estimates of absolute cell counts.

Case 1: all cell types in the tissue are modeled in the cell profile matrix: Here, we estimate the number of each cell type in a region by the product of the nucleus count in the region and the cell type's estimated proportion in the region:

$$\text{estimated cell counts} = \text{nuclei} \cdot \hat{\beta}/\textstyle\sum\hat{\beta}. \tag{1}$$

Case 2: the tissue contains cell types that are not modeled by the cell profile matrix. The motivating case here is immune deconvolution in tumors, where

cancer cell profiles are often omitted from the model. If at least one profiled region consists of entirely cells modeled by the cell profile matrix, then call the sum of its cell abundance scores $\beta_{\max}$. Then for all regions, take

$$\text{estimated cell counts} = \text{nuclei} \cdot \hat{\beta}/\beta_{\max}. \tag{2}$$

**Analysis of cell pellet array study.** Genes for normalization were selected by applying the geNorm algorithm[48] to the 50 highest-expressing genes, with lower expressing genes not considered in order to minimize the impact of system background on normalization. Each segment's expression profile was normalized using the geometric mean of the resulting 27 reference genes. The expression profiles of the pure cell lines were estimated using the median expression of the 4 unmixed replicates from each cell line. These two profiles were then scaled to have the same median expression level.

Log-normal deconvolution was run using the log-normal deconvolution algorithm. Non-negative least squared deconvolution was run by taking

$$\hat{\beta} = \mathrm{argmin}_\beta ||Y - (B + X\beta)||, \tag{3}$$

subject to the constraint that $\beta \geq 0$. Optimization was performed using the R function optim. The background term $B$ was included because ignoring background would disadvantage NNLS in the comparison. Nu support vector regression was run using svm function from R package e1071, with $\nu$ set to 0.75, a linear kernel, and without scaling. v-SVR does not allow for explicit modeling of background signal, so normalized expression data was background-subtracted before entry into v-SVR. DWLS was run using code from https://github.com/dtsoucas/DWLS.

To compute genes' influence, deconvolution was run once with the complete gene set and once with each gene omitted. Each gene's influence was reported as the absolute value difference in estimated HEK293T proportion between deconvolution with the complete gene set and deconvolution with the leave-one-out gene set.

**Screening for genes suitable for tumor-immune deconvolution.** Each TCGA sample was scored for abundance of diverse immune and stromal cells using the geometric mean of previously reported marker genes[11,12]. Then, in each cancer

type, we used log-normal regression to model each gene as follows:

$$\log(\mathbf{y}) = \log\Big(\beta_0 + \mathbf{X}_{\text{Tcell}}\beta_{\text{Tcell}} + \mathbf{X}_{\text{Bcell}}\beta_{\text{Bcell}} + \mathbf{X}_{\text{macrophage}}\beta_{\text{macrophage}} + \dots\Big) + \boldsymbol{\varepsilon},$$

$$(4)$$

where y is the vector of the gene's (linear-scale) expression across all samples in a cancer type, $\mathbf{X}_{\text{cell}}$ is the vector of a cell type's estimated abundance across all samples, and $\varepsilon$ is a vector of normally-distributed noise. We constrained all $\beta$ terms to be ≥0. The use of log-normal regression was motivated by the same considerations used in our deconvolution method: expression from mixed cell types compounds additively[49], but noise in gene expression is more Gaussian and homoscedastic after log-transformation (Supplementary Figs. 1 and 2).

In this model, the $\beta_0$ term represents the gene's average expression in a tumor when no immune cells are present. Then we can measure a gene's proportion of tumor-intrinsic expression with $\beta_0/\text{mean}(\mathbf{y})$. Ideal genes for deconvolution will have $\beta_0/\text{mean}(\mathbf{y})$ very close to 0; genes with substantial contamination from cancer cells will have $\beta_0/\text{mean}(\mathbf{y})$ near 1.

**Derivation of the SafeTME cell profile matrix.** Three datasets were used to define the SafeTME cell profile matrix for deconvolution of the tumor microenvironment: expression profiles of flow-sorted PBMCs for use in deconvolution of blood samples[5], scRNA-seq of finely clustered immune cell types[17], and RNA-seq profiles of 6 cell populations flow-sorted from lung tumors[18].

Cell-type profiles from PBMCs were used whenever possible, since flow-sorting on surface markers is the gold standard for classifying immune cells. Profiles were taken for naive B cells, memory B cells, plasmablasts, naive CD4 T cells, memory CD4 T cells, naive CD8 T cells, memory CD8 T cells, T-regulatory cells, NK cells, plasmacytoid DCs, myeloid DCs, conventional monocytes, non-conventional/intermediate monocytes, and neutrophils. We omitted profiles of PMBC cell populations expected to be vanishingly infrequent in tumors: basophils, MAIT cells, and T gamma delta cells.

From the tumor scRNA-seq dataset, we took the profiles for macrophages and mast cells, which are not present in PBMCs. We defined a mast cell profile as the average of the two reported mast cell clusters' profiles, and the macrophage profile as the average of the nine reported macrophage cluster profiles[17]. The mast cell profile was scaled to have the same 80th percentile and the average 80th percentile of the PMBC cell profiles; the macrophage profile was scaled to have the same 80th percentile as the PBMC conventional monocytes profile. The 80th percentile was chosen to negotiate two competing criteria: the need to use a quantile high enough to fall within the set of robustly expressed genes, and the need to use a quantile low enough to be statistically stable. This choice of a scaling factor is necessarily arbitrary, and error in its selection causes bias when comparing the abundance of one cell type to another. If we had chosen 0.7, the SafeTME would return mast cell abundances 40% higher and macrophage abundances 120% higher. If we had chosen 0.9, we would see mast cells 43% lower and macrophages 23% lower. Due to the uncertainty in scaling different cell type's profiles, comparison of one cell-type's abundance to another's is always fraught; the need to compute mast cell and macrophage scaling factors across datasets makes their relative abundance estimates subject to additional error.

From the flow-sorted lung tumor dataset[18], we derived profiles for endothelial cells and fibroblasts. Four endothelial cell samples with low signal were removed, as were 8 fibroblast samples with low signal. The remaining replicate samples were normalized using their 90th percentiles. In this dataset, the 90th percentile fell within the robustly expressed genes, exceeding 2 TPM in every sample, but left over a thousand genes above it to avoid excessive influence by noise in a small number of extreme high expressers. For the purposes of deriving an average profile, any standard RNA-seq normalization method would have sufficed. Endothelial cell and fibroblast profiles were defined by the median expression profiles of their replicate samples. We chose the median to represent a gene's central tendency across replicate samples because the mean was potentially too impacted by high outlier expression values, and the geometric mean was inappropriate for data with expression values of zero. The cell profiles extracted from the three datasets then were combined into a single matrix, which was reduced to a subset of 1180 highly informative genes.

It is inevitable that the combined matrix contains numerous systematic biases, such as platform effects, noise in the experimental results of the original cell profile matrices, and gene expression differences in blood versus tumor. To reduce these effects, we employed the following procedure. First, we performed deconvolution on three TCGA datasets: colon adenocarcinoma, lung adenocarcinoma, and melanoma. Most genes were consistently over- or underestimated by the deconvolution fits, and these biases were consistent across datasets. Each gene's bias was estimated with the geometric mean of the ratios between its observed expression values and its predicted expression values from the deconvolution fits. Finally, each gene's row in the cell profile matrix was then rescaled by its expected bias.

We removed genes estimated by our TCGA analysis to have more than 20% of transcripts derived from tumor cells. The final SafeTME cell profile matrix, from 18 cell types and 906 genes, is reported in the Supplementary Data.

**Derivation of profile matrices from public scRNA-seq data.** Cell profile matrices were generated using published single-cell RNA-seq datasets from human[19–33] and mouse[34–37]. In all datasets, each cell's cell type was reported by the original authors. When possible, we used the normalized data from the original publications. When only raw data was available, cells were normalized to total counts. These datasets were filtered for cells with at least one count from at least 100 different genes. Cell profiles were only calculated for cell types with 15 or more cells. Cell-type profiles were created by taking the average expression of each gene across all cells belonging to the cell type. The gene list was subset for genes that were expressed in at least one cell type and that are present in NanoString's GeoMx Human Whole Transcriptome Atlas or Mouse Whole Transcriptome Atlas panels, or, in for the SARS-CoV-2-infected tissues, from the GeoMx COVID-19 Immune Response Atlas.

**Capturing central tendencies.** Both arithmetic means and geometric means are used throughout this work. Our rule for choosing a summary statistic for the central tendency of a variable was as follows: We used geometric means for ratios and for highly right-skewed data with relatively few zeroes, including TCGA RNA-seq data and lower-thresholded GeoMx data. We used arithmetic means for naturally linear-scale data, e.g., correlations or cell counts; and for data with abundant zeroes, e.g., scRNA-seq results. In one case of right-skewed data that included many zeroes, we used the median.

**Protein slide preparation.** For GeoMx DSP slide preparation, we followed GeoMx DSP slide prep user manual (MAN-10087-04). In all, 5 µm FFPE microtome sections of non-small-cell-lung cancers (NSCLC) (ProteoGenex) or cell pellet arrays (Acepix Biosciences, Inc.) were mounted onto SuperFrost Plus slides (Fisher Scientific, 12-550-15) and air-dried overnight. Slides were prepared by baking in a drying oven at 60 °C for 1 h; then the paraffin was removed with CitriSolv (Fisher Scientific, 04-355-121). The samples were rehydrated in an ethanol gradient and final wash in DEPC-treated water (ThermoFisher, AM992). Target retrieval was performed by placing slides in staining jars containing 1× citrate buffer pH 6 (Sigma Aldrich SKU C9999-1000ML) and heated in a pressure cooker on the high-temperature setting for 15 min. Slides were allowed to cool to room temperature and blocked at room temperature for an hour with Buffer W (NanoString Technologies). The primary antibody mix was made by combining the detection antibody modules (NanoString Technologies) at 1:25 and the visualization markers in Buffer W. The NSCLC were visualized with CD3-647 at 1:400 (Abcam, ab196147), CD45-594 at 1:40 (NanoString Technologies) and PanCK-532 at 1:40 (NanoString Technologies). Slides were incubated overnight at 4 °C. Slides were fixed with 4% paraformaldehyde (Thermo Scientific 28908) and the nuclei were stained with SYTO 13 (Thermo Scientific S7575) at 1:10 for 15 min.

**RNA/NGS slide preparation.** For GeoMx DSP slide preparation, we followed GeoMx DSP slide prep user manual (MAN-10087-04). In all, 5 µm FFPE microtome sections of both non-small-cell-lung cancers (NSCLC) (ProteoGenex) were mounted onto SuperFrost Plus slides (Fisher Scientific, 12-550-15) and air-dried overnight. Slides were prepared by baking in a drying oven at 60 °C for 1 h. Slides were then processed with a Leica Biosystems BOND RXm (Leica Biosystems) as specified by the NanoString GeoMx DSP Slide Preparation User Manual (NanoString Technologies, MAN-10087). Briefly, slides were processed with the Staining protocol "*GeoMx RNA DSP slide prep", the Preparation protocol "*Bake and Dewax", HIER protocol "*HIER 20 min with ER2 @ 100 °C, and Enzyme protocol "*Enzyme 1 for 15 min". For Enzyme 1 a 1 µg/mL concentration of Proteinase K (Ambion, 2546) was used. This program included target retrieval, Proteinase K digestion, and post fixation. Once the Leica run had finished slides were immediate removed and placed in 1× PBS. One at a time, slides were placed in a prepared HybEZ Slide Rack in a HybEZ Humidity Control Tray (ADC Bio, 310012) with Kimwipes damped with 2× SSC lining the bottom. In total, 200 µL of a custom RNA probe Mix at a concentration of 4 nM per probe in 1× Buffer R (NanoString Technologies), was applied to each slide. A Hybridslip (Grace Biolabs, 714022) was immediate applied over each sample. Slides were incubated in a HybEZ over (ACDBio 321720) at 37 °C for 16–24 h. After hybridization slides were briefly dipped into a 2× SSC + 0.1% Tween-20 (Teknova, T0710) to allow the coverslips to slide off then washed twice into a 2× SSC/50% formamide (ThermoFisher AM9342) solution at 37 °C for 25 min each, followed by two washes in 2× SSC for 5 min each at room temperature. Slides were then blocked in Buffer W (NanoString Technologies) at room temperature for 30 min. In total, 200 µL of a morphology marker mix was them applied to each sample for 1 h. The tumors were visualized with CD3-647 at 1:400 (Abcam, ab196147), CD45-594 at 1:10 (NanoString Technologies), PanCK-532 at 1:20 (NanoString Technologies) and SYTO 13 at 1:10 (Thermo Scientific S7575).

**GeoMx DSP sample collection.** For GeoMx DSP sample collection, we followed GeoMx DSP instrument user manual (MAN-10088-03). Briefly, tissue slides were loaded to GeoMx DSP instrument and then scanned to visualize whole tissue images. For cell pellet array samples, 300 µm ROIs in diameter were placed. For each tissue sample, we placed ROIs and segmented into two regions: PanCK-high tumor region and PanCK-low TME regions.

**GeoMx DSP NGS library preparation and sequencing**. Each GeoMx DSP sample was uniquely indexed using Illumina's i5 × i7 dual-indexing system. In all, 4 μL of a GeoMx DSP sample was used in a PCR reaction with 1 μM of i5 primer, 1 μM i7 primer, and 1× NSTG PCR Master Mix. Thermocycler conditions were 37 °C for 30 min, 50 °C for 10 min, 95 °C for 3 min, 18 cycles of 95 °C for 15 s, 65 °C for 60 s, 68 °C for 30 s, and final extension of 68 °C for 5 min. PCR reactions were purified with two rounds of AMPure XP beads (Beckman Coulter) at 1.2× bead-to-sample ratio. Libraries were paired-end sequenced (2 × 75) on a NextSeq550 up to 400 M total aligned reads.

**Pre-processing of raw GeoMx data**. GeoMx RNA data were collected using development pipelines of the commercial GeoMx Data Analysis software. GeoMx protein data were collected using nSolver (NanoString Technologies).

**Analysis of GeoMx protein and RNA benchmarking data**. Twelve segments with very low signal in either the protein or RNA results were excluded. The protein assay data were normalized with the geometric mean of the negative control antibodies, and the RNA data were normalized with the geometric mean of the negative control probes. Prior to deconvolution, the unknown cell-types algorithm was used to append tumor-specific profiles to the SafeTME matrix. Deconvolution was run using the resulting profile matrix and the SpatialDecon algorithm.

In the benchmarking analysis, all algorithms were run using the SafeTME matrix. The NNLS, v-SVR and DWLS algorithms were all run on background-subtracted data. The stereoscope requires raw count data and so was run without background subtraction. Stereoscope also requires overdispersion (logit) parameters for all genes; as these are not available for the SafeTME, we assigned all genes a logit value of −0.8, based on the average logit seen in Stereoscope's example data. In a second Stereoscope run, the SafeTME was replaced with parameters derived from lung tumor scRNA-seq data; these parameters were downloaded from https://github.com/almaan/stereoscope/blob/master/data/params-lc.zip.

**Analysis of GeoMx RNA data from a grid over a NSCLC tumor**. Raw counts from each gene in each tissue region were extracted from the NanoString GeoMx NGS processing pipeline. For each region, the expected background for each gene was estimated with the mean of the panel's 100 negative control probes. Each region's signal strength was measured with the 85th percentile of its expression vector. Three PanCK+ regions with outlier low signal strength were removed from the analysis. Each region's data was normalized with the signal-to-background method, scaling each region such that its negative control probe mean was 1. (This method is one of the manufacturer's recommended approaches for normalizing GeoMx data; it is successful because the negative control probes respond to technical factors like region size, region-specific RNA binding efficiency, and region-specific density of material to which oligos might bind.)

Prior to deconvolution, the study's pure PanCK+/tumor segments were input into the unknown cell types algorithm, resulting in ten tumor-specific expression profiles. These profiles were then appended to the SafeTME profile matrix. Deconvolution was then run using the SpatialDecon algorithm.

To derive microenvironment subtypes, clustering was performed on the matrix of estimated cell counts using the R library pheatmap.

**Calculation of residuals from cell scores in NSCLC study**. Reverse deconvolution was run as follows. Cell abundance estimates were taken from the SpatialDecon run described above. In only the stroma segments, each gene's linear-scale expression was predicted from the cell abundance estimates using the R library logNormReg[10]. An intercept term was included in the fit, and all estimates were constrained to be non-negative.

Reverse deconvolution residuals were calculated for each gene as the log2 fold change between observed expression and fitted expression, with both terms thresholded below at 1. That is, if y.observed is a gene's normalized expression, and y.fitted is its predicted expression based on the reverse deconvolution fit, then

$$\text{residuals} = \log2(\max(\text{y.observed}, 1)) - \log2(\max(\text{y.fitted}, 1)). \quad (5)$$

The following metrics were used to measure genes' dependency on cell mixing. The correlation was calculated as cor(y.observed, y.fitted). Residual SD was calculated as sd(residuals).

To identify clusters of co-expressed genes, the correlation matrix of all genes' reverse deconvolution residuals was clustered using the R function hclust. Gene modules were identified by applying the R function cutree to the resulting hierarchical clustering results.

**Reporting summary**. Further information on research design is available in the Nature Research Reporting Summary linked to this article.

## Data availability

All data from this study is available online without restriction. The library of cell profile matrices is available at https://github.com/Nanostring-Biostats/CellProfileLibrary. The in-situ benchmarking dataset is available at https://github.com/Nanostring-Biostats/ImmuneDeconBenchmark. The data used to produce this manuscript's results are available at https://github.com/Nanostring-Biostats/SpatialDecon-manuscript-analyses. The raw GeoMx gene expression datasets generated in this study are available in the Gene Expression Omnibus under accession numbers GSE174743 (benchmarking data from 5 NSCLC tumors from Fig. 4), GSE174746 (cell pellet array from Fig. 2), and GSE174749 (NSCLC tumor from Figs. 5 and 6). The datasets used to derive the SafeTME matrix are available in the Gene Expression Omnibus under accession numbers GSE127465 (lung tumor scRNA-seq), GSE107011 (sorted PBMCs RNAseq), and GSE111907 (stroma cells RNAseq). TCGA data was accessed through https://gdac.broadinstitute.org/. The HER2+ breast cancer dataset was accessed at https://github.com/almaan/her2st. The ovarian cancer dataset was accessed at https://www.10xgenomics.com/resources/datasets. Source data are provided with this paper.

## Code availability

SpatialDecon, an R library implementing these methods, is available at https://github.com/Nanostring-Biostats/SpatialDecon and on BioConductor at https://bioconductor.org/packages/release/bioc/html/SpatialDecon.html[50]. The code used to produce this manuscript's results can be found at https://github.com/Nanostring-Biostats/SpatialDecon-manuscript-analyses.

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

## Acknowledgements

We wish to acknowledge the anonymous patients who donated the tissues used in this study.

## Author contributions

P.D. implemented the statistical methods, analyzed data, and wrote the manuscript. Y.K. designed and performed wet-lab experiments and advised in interpreting results. B.N. performed wet-lab experiments. M.G. created the library of cell profile matrices. Z.Y. ran the pan-cancer analysis used to create the SafeTME matrix. E.P. worked on gene selection and panel design. J.M.B. conceived the project and advised throughout.

## Competing interests

All authors were employees and shareholders of NanoString Technologies while performing this work. No authors had nonfinancial competing interests. J.M.B. is listed on the relevant U.S. patent 10640816, "Simultaneous quantification of gene expression in a user-defined region of a cross-sectioned tissue".
