## [Peer review file · Nature Communications]

REVIEWER COMMENTS

Reviewer #1 (Remarks to the Author): Expert in multi-omics, spatial genomics, and cancer genomics

In this manuscript, the authors developed SpatialDecon, an algorithm to quantify cell type populations based on spatially resolved transcriptomics data. Experimental evaluations revealed that the log-normal regression model utilized by SpatialDecon showcased drastic improvements over existing least-squares methods to deconvolve both spatial transcriptomics and bulk gene expression data. Additionally, the authors identified seven microenvironment subtypes within one non-small-cell-lung cancer tumor based on the estimated cell-type abundance. The reviewer downloaded the R package and successfully reproduced the Vignette's result, including multiple visualizations (heatmap, pie chart, and barplot). However, a lot of scientific issues remain in terms of the mathematical modeling, algorithm design, performance evaluation, and downstream analysis interpretations.

Methods:

1. The model of lognormal regression is similar to that of non-negative least squared deconvolution except for the log-transformed variability. This transformation was motivated by the hypothesis "expression from mixed cell types compounds additively, but noise in gene expression is log-scale". However, both the assumptions "linear additivity" and "log-scale variability" lack solid preliminary data or experimental validation.
2. In the reviewer's point of view, the definition of the L2 norm in Algorithm 1 is wrong. As known, the L2 norm is defined as the square root of the sum of the squares of the values in each dimension of a vector. As to a matrix, it is defined as the square root of the largest eigenvalue of the product matrix between the matrix and its conjugate transpose matrix. Besides, the authors did not indicate whether x refers to a vector or a matrix. Moreover, the algorithms did not indicate which variables will be output finally.
3. In the first step of Algorithm 1, the threshold operation transforms all values smaller than epsilon into epsilon. Please describe the rationale. Besides, it is also not clear how to determine the optimal value of epsilon.
4. In the second step of Algorithm 1, the beta matrix was optimized in a column-wise manner separately. This relies on a hypothesis that the columns are independent with each other. However, the authors have not made such a hypothesis in this manuscript, nor do they provide relevant references to explain the underlying rationale.
5. In the fourth step of Algorithm 1, the p-value was calculated on the basis of t-distribution. Please provide appropriate supporting information or cite related references to support this hypothesis of distribution.
6. As stated by the authors, Algorithm 2, a.k.a., SpatialDecon, is designed based on Algorithm 1 plus outlier removal. It is recommended to compare the performance of the two algorithms to investigate whether the outlier removal operation could obviously improve the performance over Algorithm 1.
7. The choice of cutoffs in this manuscript seems random rather than possessing solid rationales, e.g., "applying the geNorm algorithm to the 50 highest-expressing genes", "Choose delta as the expression level below" in Algorithm 2, "For all $\{i, j\}$ with $|R_{i,j}| > 3$ " in Algorithm 2, "Collapse the segments into 10 clusters" in Algorithm 3, and "using the geometric mean of the resulting 27 reference genes". A detailed justification of these cutoffs is an important part of scientific rigor.
8. Both arithmetic and geometric means were frequently used throughout the algorithms. It is necessary to explain the underlying rationales for the choice of arithmetic and geometric means theoretically or experimentally.

Results:

1. The authors used 700 um diameter circular region as spots to generate sequencing and regarded as a known control sample to evaluate four algorithms' performance. However, in the real spatial transcriptomics data, the spot was usually not that large. For example, slide-seq's resolution is 10um, and 10x Visium is 50 um. Therefore, the author should provide additional results to validate the heterogeneity difference between 700 um and 50 um. Furthermore, the author used 700 um to mimic

spot level data, and the software might not have sufficient evidence to support the generalizability of SpatialDecon.

2. At the end of the paper, the author mentioned that SpatialDecon can reveal co-regulated gene modules. In the reviewer's understanding, co-regulated gene modules are potentially sharing the same cis-regulatory elements. However, the author did not provide evidence to validate the potential cis-regulatory elements for identified co-regulated modules.

3. For benchmarking tools comparison, CIBRTSORT+ should be included.

4. For the result of using tumor microenvironment, it is good to identify different immune cell populations (myeloid cell, B cell, CD4 T cell, and CD8 T cell). However, in the real tumor microenvironment, there should be more specific cell populations. For example, CD8 T cells have the subset of effector, exhausted progenitor CD8 T cell, exhausted CD8 T cell. Therefore, the algorithm did not show a significant power in revealing the high-resolution for specific immune cell populations.

Codes:

1. The reviewer understands that using formulated mocking data is to save running time. However, it is worthy of putting a real large dataset as an example. Furthermore, the author should give a table of computing performance on different datasets, including execution time and actual memory cost.

2. If a user wants to upload 10X Visium spatial data, is there any portal to load in? If SpatialDecon cannot receive other data types, the generalizability might be limited.

3. The visualization function (for example, the sector distribution chart) might be packaged and form an individual function so that it can be easily used.

4. It will be interesting to have a connection between the proposed tool with the existing analysis tools. For example, if there is a functional can extract the cell abundance score and add it to the Seurat object as a metafile, it will be very helpful.

5. For the reverse deconvolution results, how can we know which deconvoluted cell populations were from the same spot?

Minor issues:

1. Some sentences were not written properly, e.g., several sentences begin with "And".

2. Some terminologies with the same meaning should be unified, e.g., "cell abundance" and "cell type abundance", "lognormal" and "log-normal", etc.

3. Some variables and terminologies were not predefined, e.g., "observation", "data point", "unmodelled cells", and variables I, J in Online Methods.

4. There is no label to indicate Figure 2a and Figure 2b in Figure 2.

5. Some grammar mistakes exist, e.g., "an algorithm based in log-normal regression".

6. Every abbreviation should be defined at its first appearance in the manuscript, e.g., non-small-cell-lung cancers (NSCLC).

7. The authors are suggested to use the appropriate professional terminology. For example, avoid using "vs." for comparison.

8. The author should improve figure quality and use larger font because some figure legend is hard to tell (e.g., Figure 5b).

Reviewer #2 (Remarks to the Author): Expert in bioinformatics and cell type deconvolution

Danaher et al. present the algorithm SpatialDecon, which produces spatially-resolved cell abundance estimates. The manuscript reads nicely and shows very relevant results that researchers working on spatial transcriptomics and deconvolution should be aware of (e.g. using log-normal regression and its robustness to high-influence marker genes) and provide a new cell profile matrix for the tumor microenvironment (SafeTME).

However, in its current form, I have several concerns which I think should be taken into consideration and commented in the manuscript:

Primary comments:

- While I am convinced with their proposed log-normal regression (greater accuracy; Sup. Fig 1) and their comparison with least-squares deconvolution, v-SVR, DWLS and Stereoscope, I would like to see, if possible (for completeness and even higher relevance), a broader comparison with three extra methodologies: the spatial version of DWLS (SpatialDWLS; bioRxiv); "SPOTlight" (seeded NMF regression; Nucleic Acids Research) and RCTD (Nature Biotechnology).

- Figure 1 shows "p-values and confidence intervals for cell estimates" as one of the advances in deconvolution algorithms. However, I did not find information about how these intervals are computed anywhere in the manuscript/methods section.

- On their Github repository (<https://github.com/Nanostring-Biostats/SpatialDecon>), I could read: Its minimal required input is:

- 1) A normalized data matrix
- 2) A matrix of expected background counts at each element of the normalized data matrix
- 3) A matrix of expected cell type expression profiles

While elements 1 and 3 are represented in Figure 1, the second one (which is crucial) is not included. Furthermore, it's written that "the expected background for each gene was estimated with the mean of the panel's 100 negative control probes". Is there more information about these negative control probes (e.g. to be able to apply SpatialDecon to other spatial technologies?) or are these specific (intellectual property) of NanoString Technologies (GeoMx protein and GeoMx RNA assays)? This key piece of information should be clearly indicated both in Figure 1 and the abstract and Discussion sections.

- In the section "Derivation of the SafeTME cell profile matrix for deconvolution in the tumor microenvironment", why mast and macrophage profiles were scaled to the same 80th percentile, endothelial and fibroblasts were defined by the median expression and the remaining samples were normalized using their 90th percentiles? This should be explained/backed-up quantitatively (e.g. a criterion used to define such percentile threshold)

- In the section "A library of scRNA-seq-derived cell profile matrices from diverse tissue types", authors should explain in more detail what they mean by "domain knowledge".

- In the section "Derivation of cell profile matrices from public scRNA-seq datasets" authors should clarify what they did exactly by "All cluster names were manually reviewed for correctness".

Minor comments:

- The use of the word "safe" in the header "A pan-cancer screen for genes with negligible expression in cancer cells identifies genes safe for immune deconvolution in tumors" is not ideal in my opinion.
- "PD invented the statistical methods". Is it really invention or implementation?

To the editors and our reviewers:

We would like to thank both reviewers for their thorough and informed readings of our manuscript and code. We have addressed every comment, in nearly all cases amending the manuscript or the SpatialDecon R library. We have added a new dataset and performed numerous new analyses, the results of which are described below and in 3 new supplemental figures. We believe these edits have strengthened the manuscript considerably.

Below, please find reviewer comments in blue and our responses in black.

Reviewer #1 (Remarks to the Author): Expert in multi-omics, spatial genomics, and cancer genomics

In this manuscript, the authors developed SpatialDecon, an algorithm to quantify cell type populations based on spatially resolved transcriptomics data. Experimental evaluations revealed that the log-normal regression model utilized by SpatialDecon showcased drastic improvements over existing least-squares methods to deconvolve both spatial transcriptomics and bulk gene expression data. Additionally, the authors identified seven microenvironment subtypes within one non-small-cell-lung cancer tumor based on the estimated cell-type abundance. The reviewer downloaded the R package and successfully reproduced the Vignette's result, including multiple visualizations (heatmap, pie chart, and barplot). However, a lot of scientific issues remain in terms of the mathematical modeling, algorithm design, performance evaluation, and downstream analysis interpretations.

Methods:

1. The model of lognormal regression is similar to that of non-negative least squared deconvolution except for the log-transformed variability. This transformation was motivated by the hypothesis "expression from mixed cell types compounds additively, but noise in gene expression is log-scale". However, both the assumptions "linear additivity" and "log-scale variability" lack solid preliminary data or experimental validation.

Good point. We have amended the manuscript to justify these choices. We summarize below:

Regarding the assumption of log-scale variability:

The manuscript is now more clear in explaining the rationale for a log-scale variance model: namely, that log-transformation largely corrects the extreme departures from normality and homoscedasticity found in linear-scale gene expression. We demonstrate this claim in Supplementary Figure 1.

Specifically, in the results, we have replaced "This approach retains the mean model of least-squares regression while modelling variability on the log-scale." with "This approach retains the mean model of least-squares regression while modelling variability on the log-scale, which largely corrects the skewness and unequal variance of gene expression data (Supplementary Figure 1)."

And in the methods, we have replaced "noise in gene expression is log-scale" with "noise in gene expression is more Gaussian and homoscedastic after log-transformation (Supplementary Figure 1)"

Regarding the assumption of linear additivity:

When we say “expression from mixed cell types compounds additively”, we simply mean that mRNA transcripts from different cell types add atop each other to produce the total transcripts observed, e.g. that a gene’s total counts = counts from T-cells + counts from B-cells + etc...

The assumption of a linear mean model has been shared by essentially all deconvolution methods since the appropriateness of this model was pointed out by Zhong & Liu (2012), “Gene expression deconvolution in linear space.” We now cite this work in the Methods when we claim “mixed cell types compounds additively”.

2. In the reviewer’s point of view, the definition of the L2 norm in Algorithm 1 is wrong. As known, the L2 norm is defined as the square root of the sum of the squares of the values in each dimension of a vector. As to a matrix, it is defined as the square root of the largest eigenvalue of the product matrix between the matrix and its conjugate transpose matrix. Besides, the authors did not indicate whether x refers to a vector or a matrix.

Good point. We have corrected our terminology to reflect that we are not using L2 norm but rather an equivalent formulation, revising the text to read, “Let $\|x\|$ denote the operator of a vector x such that $\|x\| = \text{mean}(x^2)$.”

Moreover, the algorithms did not indicate which variables will be output finally.

Good point. We have expanded on the algorithm descriptions to do so.

3. In the first step of Algorithm 1, the threshold operation transforms all values smaller than epsilon into epsilon. Please describe the rationale.

This threshold operation is necessary to avoid negative infinity values resulting from log-transformed zeroes. We have amended the Algorithm 1 text in the Methods to explain this rationale, saying “To avoid negative-infinity values when log-transforming zero-valued elements of Y .”

Besides, it is also not clear how to determine the optimal value of epsilon.

We use epsilon equal to the minimum non-zero value in Y . This somewhat conservative choice was made to alter the data as lightly as possible. It is likely that a higher value of epsilon would be more optimal in some datasets, but in other datasets a higher epsilon would conceal meaningful signal. In early explorations with this method, different choices of epsilon did not meaningfully change the algorithm’s output.

4. In the second step of Algorithm 1, the beta matrix was optimized in a column-wise manner separately. This relies on a hypothesis that the columns are independent with each other. However, the authors have not made such a hypothesis in this manuscript, nor do they provide relevant references to explain the underlying rationale.

Good point: we see how this could cause reader confusion. We now state in the text that “Separate optimization over columns of Y is permissible since different columns of Y represent data from different region of a tissue, and so can be treated as effectively independent.”

5. In the fourth step of Algorithm 1, the p-value was calculated on the basis of t-distribution. Please provide appropriate supporting information or cite related references to support this hypothesis of distribution.

Good catch. The claim of a t-distribution was not correct; it should have been a standard normal distribution. The theory behind this p-value is the Wald test, which states that a maximum likelihood estimate (in our case, the estimate that minimizes squared log-scale errors) is asymptotically normally distributed.

In practice, the distinction between the t and standard normal distributions is negligible, since high numbers of genes lead to a t-distribution with high degrees of freedom, which is very close to a normal distribution. Correcting the t CDF to a normal CDF leads to slightly smaller p-values.

We have corrected the text in the Methods section to state standard normal instead of t, and we have updated the spatialdecon library to use the Z distribution instead of the t distribution.

6. As stated by the authors, Algorithm 2, a.k.a., SpatialDecon, is designed based on Algorithm 1 plus outlier removal. It is recommended to compare the performance of the two algorithms to investigate whether the outlier removal operation could obviously improve the performance over Algorithm 1.

Good idea. Using the mRNA and protein serial sections validation dataset from Figure 4, we compared spatialdecon with and without outlier removal. The results were nearly identical, with average correlations achieved without outlier removal beating average correlations with outlier removal by 0.008 (paired t-test $p = 0.16$).

Given the equivalence seen above, and under the guiding principle of “avoid returning badly wrong results”, we propose to continue defaulting to removing outliers. But we have now updated the spatialdecon R package to let the user disable this option.

7. The choice of cutoffs in this manuscript seems random rather than possessing solid rationales, e.g., “applying the geNorm algorithm to the 50 highest-expressing genes”, “Choose delta as the expression level below” in Algorithm 2, “For all $\{i, j\}$ with $|R_{i,j}| > 3$ ” in Algorithm 2, “Collapse the segments into 10 clusters” in Algorithm 3, and “using the geometric mean of the resulting 27 reference genes”. A detailed justification of these cutoffs is an important part of scientific rigor.

Good point. We now detail these choices further in the manuscript as detailed below:

i. Re: “applying the geNorm algorithm to the 50 highest-expressing genes” and “using the geometric mean of the resulting 27 reference genes”:

The manuscript now states, “Genes for normalization were selected by applying the geNorm algorithm to the 50 highest-expressing genes, excluding lower genes to minimize the impact of system background on normalization.”

The choice of looking at exactly the 50 highest genes was arbitrary but reasonable. To confirm that other reasonable normalization options produce very similar results, we ran geNorm on the

top 50, top 100, and top 200 genes. The correlations between the resulting (log-scale) normalization factors were all > 0.994 , suggesting this arbitrary choice does not impact the analysis.

ii. Re: "Choose delta as the expression level below" in Algorithm 2

We had not sufficiently studied the choice of delta. Inspired by this comment, we analyzed the sensitivity of the deconvolution results to the choice of delta. Again using our protein-RNA serial sections validation study, we ran spatialdecon with delta values of 0.01, 0.5 (our recommended default when analyzing signal-to-background-scale data), 1, and 2. We found negligible impact on performance, with mean correlation to protein changing by less than 0.005 across all settings.

iii. Re: "For all $\{i, j\}$ with $|R_{i,j}| > 3$ " in Algorithm 2:

This refers to the setting used in outlier removal, which we have addressed in our response to question 6.

iv. Re: "Collapse the segments into 10 clusters" in Algorithm 3:

There is no clear universal solution to choosing this parameter. To better guide users, we have added the following text to the manuscript:

"Collapse the segments into G clusters by applying the R functions hclust and cutree to their log-transformed expression profiles. The value of G should be set large enough to capture the diversity of the segments of the unmodelled cell type and small enough to avoid adding excessive terms to the deconvolution model. For healthy cell types with homogeneous behavior, 1-2 clusters is adequate; for tumor cells, which can be highly heterogeneous, a higher G is recommended. We use $G=10$ in the tumor analyses in this manuscript."

8. Both arithmetic and geometric means were frequently used throughout the algorithms. It is necessary to explain the underlying rationales for the choice of arithmetic and geometric means theoretically or experimentally.

Good point. Our general operating principle was as follows:

We use geometric means for 1. highly right-skewed data with relatively few zeroes, including TCGA RNA-seq data and lower-thresholded GeoMx data, and 2. ratios.

We use arithmetic means for 1. naturally linear-scale data, e.g. correlations or cell counts, and 2. data with abundant zeroes, e.g. scRNA-seq results.

This logic is explained in the new Methods section, "Capturing central tendencies"

Results:

1. The authors used 700 um diameter circular region as spots to generate sequencing and regarded as a known control sample to evaluated four algorithms' performance. However, in the real spatial

transcriptomics data, the spot was usually not that large. For example, slide-seq's resolution is 10um, and 10x Visium is 50 um. Therefore, the author should provide additional results to validate the heterogeneity difference between 700 um and 50 um. Furthermore, the author used 700 um to mimic spot level data, and the software might not have sufficient evidence to support the generalizability of SpatialDecon.

This raises a key point unaddressed in the original submission: how is spatialdecon expected to perform in small regions like those of Visium experiments? To address this question, we ran immune deconvolution in a colorectal cancer, profiling regions ranging from 1119 to 145633 square microns, including 15 regions smaller than the 2375 square microns of a 55-um diameter spot. The estimated immune composition in Visium-sized regions was effectively identical to that in larger regions. Three of the smallest regions (1100um² – 170um²) returned slightly different proportions of the same cell types found in larger regions. We have added this demonstration to our supplementary material.

2. At the end of the paper, the author mentioned that SpatialDecon can reveal co-regulated gene modules. In the reviewer's understanding, co-regulated gene modules are potentially sharing the same cis-regulatory elements. However, the author did not provide evidence to validate the potential cis-regulatory elements for identified co-regulated modules.

Good point. Our terminology was misleading. Our use of "co-regulated" was meant to describe genes with correlated expression within a cell population, with correlation potentially but not necessarily arising from cis-regulatory elements. We have revised the manuscript to make this clear, and now write "co-expressed" instead of "co-regulated".

That said, the question of cis-regulatory elements is an interesting one. Unfortunately, the gene clusters we identified were too small for accurate discovery of conserved regulatory elements. We attempted conserved motif discovery using Homer (<http://homer.ucsd.edu/homer>) and oPOSSUM (<http://opossum.cisreg.ca/cgi-bin/oPOSSUM3/>). Neither method revealed strong results, despite the fact that known regulation exists for MHC Class II, as we now note in the manuscript: "It has been previously shown that these HLA genes share regulatory elements".

3. For benchmarking tools comparison, CIBERSORT+ should be included.

This was a good suggestion. We contacted Cibermed Inc, which runs CIBERSORTx as a consulting service. (CIBERSORTx is not open-source software.) They replied that our benchmarking dataset has too few genes from their cell profile matrix (the "LM22" matrix), and so CIBERSORTx could not be appropriately evaluated in our dataset. They declined to make their algorithms available for use with other matrices.

Our benchmarking dataset's use of just 1700 genes limits its use in evaluating competing cell profile matrices. However, it still serves to contrast deconvolution algorithms. Our inclusion of v-SVR in our benchmarking exercise captures the core of the CIBERSORT algorithm. CIBERSORTx introduces additional bells and whistles which cannot be easily recapitulated.

4. For the result of using tumor microenvironment, it is good to identify different immune cell populations (myeloid cell, B cell, CD4 T cell, and CD8 T cell). However, in the real tumor

microenvironment, there should be more specific cell populations. For example, CD8 T cells have the subset of effector, exhausted progenitor CD8 T cell, exhausted CD8 T cell. Therefore, the algorithm did not show a significant power in revealing the high-resolution for specific immune cell populations.

This is an important point neglected in our first version. Although the submitted manuscript only showed somewhat coarsely-defined cell types, this limitation results from the safeTME's reliance on Monaco (2019), not on an intrinsic limitation of the algorithm. In other studies, we have found that spatialdecon still works when given a very fine-grained cell profile matrix with >40 cell types.

To demonstrate this point, we assembled a matrix of 44 cell types by combining our fibroblast and endothelial cell profiles with 42 immune cell subtypes reported from a tumor scRNA-seq study (Zilionis 2019). As an example of its granularity, this matrix contains 7 T-cell subtypes and 9 macrophage subtypes.

Although we know of no data that can validate results for any single cell subcluster, we used our benchmarking study to confirm the basic sanity of the algorithm when using granular cell profiles. After performing deconvolution with the 44 cell type matrix, we added the scores from related cell types for comparison to protein. E.g., we compared the total abundance of the 9 macrophage cell types to CD68 protein expression. SpatialDecon's concordance with marker proteins was similarly accurate as we reported with the SafeTME matrix. We now show these results in the supplement and refer to them in the results.

We also applied this more granular cell profile matrix to the tumor shown in Figures 5 & 6. Cell subsets showed distinct and internally consistent spatial distributions. For example, the cycling T-cell cluster, hT7, was primarily present in the B-cell enriched regions, which are likely tertiary lymphoid structures. In contrast, cluster hT5 was almost exclusively found in the upper-right corner of the tumor, away from the B-cells. We now show these results in the supplement and allude to them in the results.

Codes:

1. The reviewer understands that using formulated mocking data is to save running time. However, it is worthy of putting a real large dataset as an example.

Good idea. The SpatialDecon R library now includes a vignette showing the use of SpatialDecon on the NSCLC tumor described in our manuscript. This update will appear after BioConductor approves our changes. Until then, it can be seen at: https://github.com/Nanostring-Biostats/SpatialDecon/blob/revision-for-journal/vignettes/SpatialDecon_vignette_NSCLC.Rmd. This vignette will generate if the package is installed from the branch "revisions-for-journal".

Furthermore, the author should give a table of computing performance on different datasets, including execution time and actual memory cost.

Good idea. We have added stats about memory usage and runtime to the package's readme.

2. If a user wants to upload 10X Visium spatial data, is there any portal to load in? If SpatialDecon cannot receive other data types, the generalizability might be limited.

The package supports Visium and GeoMx with equal convenience. For either platform, the user need only extract 1. An expression matrix, and 2. The estimated background level in each sampled region. (In Visium, as discussed in more detail after a question from Reviewer 2, expected background is just 0.)

3. The visualization function (for example, the sector distribution chart) might be packaged and form an individual function so that it can be easily used.

We appreciate this feedback, and we are already working on a larger visualization package that includes the function alluded to above.

4. It will be interesting to have a connection between the proposed tool with the existing analysis tools. For example, if there is a functional can extract the cell abundance score and add it to the Seurat object as a metafile, it will be very helpful.

Good idea. We have added an S4 method to our package that facilitate applying spatialdecon to Seurat S4 objects. This update will appear after BioConductor approves our changes. Until then, you can see the new code at <https://github.com/Nanostring-Biostats/SpatialDecon/blob/revision-for-journal/R/runspatialdecon.R>.

5. For the reverse deconvolution results, how can we know which deconvoluted cell populations were from the same spot?

Good question. The spatialdecon algorithm outputs spot-by-spot estimates of cell abundance, which, along with spot-by-spot gene expression, is all that is needed for reverse deconvolution to be run.

Minor issues:

1. Some sentences were not written properly, e.g., several sentences begin with “And”.

We have modified these sentences.

2. Some terminologies with the same meaning should be unified, e.g., “cell abundance” and “cell type abundance”, “lognormal” and “log-normal”, etc.

Good point. We have committed the manuscript to “log-normal” and “cell abundance”.

3. Some variables and terminologies were not predefined, e.g., “observation”, “data point”, “unmodelled cells”, and variables I, J in Online Methods.

Good point. We have now defined all these terms at the beginning of the methods sections where they first appear. The term “data point” we have replaced with more precise language.

4. There is no label to indicate Figure 2a and Figure 2b in Figure 2.

Good catch. We have corrected this.

5. Some grammar mistakes exist, e.g., “an algorithm based in log-normal regression”.

Good catch. We have corrected this oversight and reviewed the manuscript for other grammatical errors.

6. Every abbreviation should be defined at its first appearance in the manuscript, e.g., non-small-cell-lung cancers (NSCLC).

We have amended this.

7. The authors are suggested to use the appropriate professional terminology. For example, avoid using “vs.” for comparison.

We have replaced “vs.” with more formal terminology wherever it appeared.

8. The author should improve figure quality and use larger font because some figure legend is hard to tell (e.g., Figure 5b).

We have amended this.

Reviewer #2 (Remarks to the Author): Expert in bioinformatics and cell type deconvolution

Danaher et al. present the algorithm SpatialDecon, which produces spatially-resolved cell abundance estimates. The manuscript reads nicely and shows very relevant results that researchers working on spatial transcriptomics and deconvolution should be aware of (e.g. using log-normal regression and its robustness to high-influence marker genes) and provide a new cell profile matrix for the tumor microenvironment (SafeTME).

However, in its current form, I have several concerns which I think should be taken into consideration and commented in the manuscript:

Primary comments:

- While I am convinced with their proposed log-normal regression (greater accuracy; Sup. Fig 1) and their comparison with least-squares deconvolution, v-SVR, DWLS and Stereoscope, I would like to see, if possible (for completeness and even higher relevance), a broader comparison with three extra methodologies: the spatial version of DWLS (SpatialDWLS; bioRxiv); "SPOTlight" (seeded NMF regression; Nucleic Acids Research) and RCTD (Nature Biotechnology).

Good suggestion. We have added SpatialDWLS results to our benchmarking study. Unfortunately, both RCTD and SPOTlight require matching single cell RNA-seq data, and so they are unable to process our benchmarking dataset.

These methods' reliance on a full scRNA-seq dataset limits their use to a subset of the datasets that spatialdecon can process. This limitation is most acute in cancer studies, where the heterogeneity of tumor gene expression means public scRNA-seq data cannot be used to deconvolve a new tumor's spatial data.

Our attempts to apply these other methods suggested to us that SpatialDecon's ability to use reference matrices rather than scRNA-seq data is an enabling feature that distinguishes it in the landscape of spatial deconvolution methods. We now comment on this in a new paragraph in the discussion.

- Figure 1 shows "p-values and confidence intervals for cell estimates" as one of the advances in deconvolution algorithms. However, I did not find information about how these intervals are computed anywhere in the manuscript/methods section.

Good catch. We have corrected this oversight and added the below text to the "algorithm2" section of the manuscript: " Calculate the 95% confidence interval for each $\beta_{i,j}$ with $\widehat{\beta}_{j,i} \pm 1.96 (\sum_{j,j}^{(1)})^{1/2}$ "

- On their Github repository (<https://github.com/Nanostring-Biostats/SpatialDecon>), I could read: Its minimal required input is:

- 1) A normalized data matrix
- 2) A matrix of expected background counts at each element of the normalized data matrix
- 3) A matrix of expected cell type expression profiles

While elements 1 and 3 are represented in Figure 1, the second one (which is crucial) is not included.

Furthermore, it's written that "the expected background for each gene was estimated with the mean of the panel's 100 negative control probes". Is there more information about these negative control probes (e.g. to be able to apply SpatialDecon to other spatial technologies?) or are these specific (intellectual property) of NanoString Technologies (GeoMx protein and GeoMx RNA assays)? This key piece of information should be clearly indicated both in Figure 1 and the abstract and Discussion sections.

Good point. We have clarified the role of background in several places in the manuscript.

To summarize: the estimation of background is a crucial but trivial step in analysis of GeoMx data. All GeoMx panels have "ERCC" negative control probes targeting sequences alien to the human genome. The counts seen from these probes measures the tendency of all probes to bind off-target and produce background counts. Thus to estimate background in any given region of a GeoMx study, we simply take the mean counts from the negative controls in that region. In contrast, because Visium does not use a hybridization-based chemistry, to our knowledge Visium data has no background.

How we've addressed this comment in the paper:

- The abstract now notes that our new algorithm explicitly models background.
- We have added information about the negative control probes and the calculation of background to the Methods, in the new section, "Estimating Background."
- In the Methods, we note that in a Visium study, if you were able estimate diffusion of location UMI's across spots, you could estimate diffusion-derived background.
- The discussion now notes the importance of modelling background.
- We would like to omit the trivial step of background estimation in Figure 1 to avoid distracting from the figure's other important content.

- In the section "Derivation of the SafeTME cell profile matrix for deconvolution in the tumor microenvironment", why mast and macrophage profiles were scaled to the same 80th percentile, endothelial and fibroblasts were defined by the median expression and the remaining samples were normalized using their 90th percentiles? This should be explained/backed-up quantitatively (e.g. a criterion used to define such percentile threshold)

This is a good question; our initial writeup was unclear about the rationale for these choices. The manuscript now explains:

- "The 80th percentile was chosen to negotiate two competing criteria: the need to use a quantile high enough to fall within the set of robustly expressed genes, and the need to use a quantile low enough to be statistically stable."
- "In this dataset, the 90th percentile fell within the robustly expressed genes but left enough genes above it that it would not be excessively influenced by noise in a small number of extreme high expressers."

- “We chose the median to represent a gene’s central tendency across replicate samples because the mean was potentially too impacted by high outlier expression values, and the geometric mean was inappropriate for data with zero expression values.”

- In the section “A library of scRNA-seq-derived cell profile matrices from diverse tissue types”, authors should explain in more detail what they mean by “domain knowledge”.

Good point. The text now states, “We named cell clusters using published marker genes, curated to better reflect the known biology of each cell type based on the scientific literature.”

- In the section “Derivation of cell profile matrices from public scRNA-seq datasets” authors should clarify what they did exactly by “All cluster names were manually reviewed for correctness”.

Good point. The text now states, “All cluster names were manually reviewed for correctness, both by comparison of PanglaoDB’s markers to the literature and by evaluating markers published in the literature but missing from PanglaoDB’s data-driven lists.”

Minor comments:

- The use of the word “safe” in the header “A pan-cancer screen for genes with negligible expression in cancer cells identifies genes safe for immune deconvolution in tumors” is not ideal in my opinion.

We have changed “safe” to “suitable”.

- “PD invented the statistical methods”. Is it really invention or implementation?

We have changed the text to read “implemented”.

REVIEWER COMMENTS

Reviewer #1 (Remarks to the Author):

Dear Authors,

Thanks for your efforts in this revision. However, I still have some concerns remaining (in terms of algorithm, evaluation, and tool development) based on the revised manuscript and response letter. I attached a PDF in the review submission since there are some figures in my responses and comments.

Best,

Reviewer #2 (Remarks to the Author):

The authors have answered all my comments and included SpatialDWLS (RCTD and SPOTlight were not included for valid reasons). While I am satisfied with many of their answers, there are still a few key aspects that, in my opinion, need improvement.

COMMENTS TO THE MANUSCRIPT

Their statement "We named cell clusters using published marker genes, curated to better reflect the known biology of each cell type based on the scientific literature" is still too vague (references?). Furthermore, even though they provide cell profile matrices and the marker genes used to classify clusters (Supp. Table 4), a clear methodology that can be reproduced is needed ("In each dataset, we derived cell clusters"... how?). Their section "Details on methods from Danaher & Kim (2020)" (<https://github.com/Nanostring-Biostats/CellProfileLibrary>) should be introduced in the manuscript (Methods section).

The scaling to the same 80th percentile, 90th or median is still not clear quantitatively (e.g. "the 90th percentile fell within the robustly expressed genes but left enough genes above it"). How many are "enough genes"? For instance, if authors require ≥ 10 genes as a threshold, this could indeed be a different percentile per cell type. This insight needs to be clarified to further investigate new datasets with their method.

Finally, during their rebuttal, the authors removed the words "in spatial gene expression data" from their section "Log-normal regression improves deconvolution performance". Since this is no longer exclusive of spatial transcriptomics, it would be great for the field if the authors analyzed and comment novelty, similarities and differences with dtangle (Hunt et al., *Bioinformatics* 2019). Specifically, with respect to this paragraph from Hunt et al.: "dtangle's approach is built on a biologically appropriate linear mixing model of linear-scale expressions but robustly fitting the model using log-transformed data and thus sets it apart from other deconvolution methods."

COMMENTS TO THE CODE

Their main command (from Github) to install the package does not work (I tried in 2 laptops, both mac OS and Linux):

COMMAND

```
devtools::install_github("Nanostring-Biostats/SpatialDecon", ref = "master", build_vignettes = TRUE)
```

```
## LOG
```

```
--- re-building 'SpatialDecon_vignette.Rmd' using rmarkdown  
Error: processing vignette 'SpatialDecon_vignette.Rmd' failed with diagnostics:  
object 'is_R_CMD_check' not found  
--- failed re-building 'SpatialDecon_vignette.Rmd'
```

```
SUMMARY: processing the following file failed:  
'SpatialDecon_vignette.Rmd'
```

```
Error: Vignette re-building failed.  
Execution halted
```

```
Error: Failed to install 'SpatialDecon' from GitHub:  
System command 'R' failed, exit status: 1, stdout + stderr (last 10 lines):  
E> --- re-building 'SpatialDecon_vignette.Rmd' using rmarkdown  
E> Error: processing vignette 'SpatialDecon_vignette.Rmd' failed with diagnostics:  
E> object 'is_R_CMD_check' not found  
E> --- failed re-building 'SpatialDecon_vignette.Rmd'  
E>  
E> SUMMARY: processing the following file failed:  
E> 'SpatialDecon_vignette.Rmd'  
E>  
E> Error: Vignette re-building failed.  
E> Execution halted
```

Furthermore, on the "Code and Software Checklist PDF (154KB)" I saw the package was already in Bioconductor with requirements $R \geq 4.0.0$. However, I tried that way too and it didn't work:

```
## COMMAND
```

```
BiocManager::install("SpatialDecon")
```

```
## LOG
```

```
Bioconductor version 3.11 (BiocManager 1.30.16), R 4.0.1 (2020-06-06)  
Installing package(s) 'SpatialDecon'  
Warning message:  
In .inet_warning(msg) :  
package 'SpatialDecon' is not available (for R version 4.0.1)
```

Nevertheless, I was able to load the different functions and data objects to test their vignette from Bioconductor (https://bioconductor.org/packages/release/bioc/vignettes/SpatialDecon/inst/doc/SpatialDecon_vignette.R) and, fortunately, that did work.

Methods:

1. The model of lognormal regression is similar to that of non-negative least squared deconvolution except for the log-transformed variability. This transformation was motivated by the hypothesis “expression from mixed cell types compounds additively, but noise in gene expression is log-scale”. However, both the assumptions “linear additivity” and “log-scale variability” lack solid preliminary data or experimental validation.

Good point. We have amended the manuscript to justify these choices. We summarize below: Regarding the assumption of log-scale variability:

The manuscript is now more clear in explaining the rationale for a log-scale variance model: namely, that log-transformation largely corrects the extreme departures from normality and homoscedasticity found in linear-scale gene expression. We demonstrate this claim in Supplementary Figure 1.

Specifically, in the results, we have replaced “This approach retains the mean model of least-squares regression while modelling variability on the log-scale.” with “This approach retains the mean model of least-squares regression while modelling variability on the log-scale, which largely corrects the skewness and unequal variance of gene expression data (Supplementary Figure 1).”

And in the methods, we have replaced “noise in gene expression is log-scale” with “noise in gene expression is more Gaussian and homoscedastic after log-transformation (Supplementary Figure 1)”

Regarding the assumption of linear additivity:

When we say “expression from mixed cell types compounds additively”, we simply mean that mRNA transcripts from different cell types add atop each other to produce the total transcripts observed, e.g. that a gene’s total counts = counts from T-cells + counts from B-cells + etc...

The assumption of a linear mean model has been shared by essentially all deconvolution methods since the appropriateness of this model was pointed out by Zhong & Liu (2012), “Gene expression deconvolution in linear space.” We now cite this work in the Methods when we claim “mixed cell types compounds additively”.

Thanks for your response. The authors only used one dataset to demonstrate the rationale for log-transformation, which is not sufficient to draw the conclusion. Please provide more data to demonstrate the claim. In addition, the distribution of noise in the data is to some extent impacted by the sequencing techniques. The deconvolution methodology in reference to Zhong & Liu (2012) was dedicated to microarray data rather than RNA-Seq or scRNA-Seq, which may not be suitable to describe the newly-developed sequencing techniques. Therefore, it is necessary to discuss whether the linear mean model is still suitable for the current sequencing data. Please clarify the rationale comprehensively via theoretical proof or references.

3. In the first step of Algorithm 1, the threshold operation transforms all values smaller than epsilon into epsilon. Please describe the rationale. Besides, it is also not clear how to determine the optimal value of epsilon.

We use epsilon equal to the minimum non-zero value in Y. This somewhat conservative choice was made to alter the data as lightly as possible. It is likely that a higher value of epsilon would be more optimal in some datasets, but in other datasets a higher epsilon would conceal

meaningful signal. In early explorations with this method, different choices of epsilon did not meaningfully change the algorithm's output.

Thanks for your response. After thresholding Y using epsilon, all the values smaller than epsilon obtain the same value, i.e., epsilon. This operation will remove variations within the data which may result from both biological or technical factors. Please perform more computational analyses to demonstrate that different values of epsilon will not drastically affect the result of Algorithm 1.

4. In the second step of Algorithm 1, the beta matrix was optimized in a column-wise manner separately. This relies on a hypothesis that the columns are independent with each other. However, the authors have not made such a hypothesis in this manuscript, nor do they provide relevant references to explain the underlying rationale.

Good point: we see how this could cause reader confusion. We now state in the text that "Separate optimization over columns of Y is permissible since different columns of Y represent data from different region of a tissue, and so can be treated as effectively independent."

Thanks for your response. However, the authors only revised this step in Algorithm 1 by adding a descriptive sentence rather than providing solid computational or reference evidence. As known, even different regions of tissue may impact each other via signaling pathways, e.g., microglia and astrocytes in the spinal cord. Hence, this statement is qualified as a rationale for step 2 in Algorithm 1.

5. In the fourth step of Algorithm 1, the p-value was calculated on the basis of t-distribution. Please provide appropriate supporting information or cite related references to support this hypothesis of distribution.

Good catch. The claim of a t-distribution was not correct; it should have been a standard normal distribution. The theory behind this p-value is the Wald test, which states that a maximum likelihood estimate (in our case, the estimate that minimizes squared log-scale errors) is asymptotically normally distributed.

In practice, the distinction between the t and standard normal distributions is negligible, since high numbers of genes lead to a t-distribution with high degrees of freedom, which is very close to a normal distribution. Correcting the t CDF to a normal CDF leads to slightly smaller p-values.

We have corrected the text in the Methods section to state standard normal instead of t, and we have updated the spatialdecon library to use the Z distribution instead of the t distribution.

Thanks for your response. Please cite relevant references regarding standard normal distribution or theoretical proof as a support of this step.

6. As stated by the authors, Algorithm 2, a.k.a., SpatialDecon, is designed based on Algorithm 1 plus outlier removal. It is recommended to compare the performance of the two algorithms to investigate whether the outlier removal operation could obviously improve the performance over Algorithm 1.

Good idea. Using the mRNA and protein serial sections validation dataset from Figure 4, we compared spatialdecon with and without outlier removal. The results were nearly identical, with

average correlations achieved without outlier removal beating average correlations with outlier removal by 0.008 (paired t-test $p = 0.16$).

Given the equivalence seen above, and under the guiding principle of “avoid returning badly wrong results”, we propose to continue defaulting to removing outliers. But we have now updated the spatialdecon R package to let the user disable this option.

Thanks for your response. Since the results are nearly identical between with and without outlier removal, I am not convinced of the performance of outlier removal in SpatialDecon. Please conduct experiments to demonstrate that the outlier removal step does decrease the outliers in data. Otherwise, this step is unnecessary to the performance of SpatialDecon. For the latter situation, please remove this step.

7. The choice of cutoffs in this manuscript seems random rather than possessing solid rationales, e.g., “applying the geNorm algorithm to the 50 highest-expressing genes”, “Choose delta as the expression level below” in Algorithm 2, “For all $\{i, j\}$ with $|R_{i,j}| > 3$ ” in Algorithm 2, “Collapse the segments into 10 clusters” in Algorithm 3, and “using the geometric mean of the resulting 27 reference genes”. A detailed justification of these cutoffs is an important part of scientific rigor.

iii. Re: “For all $\{i, j\}$ with $|R_{i,j}| > 3$ ” in Algorithm 2:

This refers to the setting used in outlier removal, which we have addressed in our response to question 6.

Thanks for your response. The authors did not describe the rationale for this operation in the response of neither question 6 nor question 7.

iv. Re: “Collapse the segments into 10 clusters” in Algorithm 3:

There is no clear universal solution to choosing this parameter. To better guide users, we have added the following text to the manuscript:

“Collapse the segments into G clusters by applying the R functions `hclust` and `cutree` to their log-transformed expression profiles. The value of G should be set large enough to capture the diversity of the segments of the unmodelled cell type and small enough to avoid adding excessive terms to the deconvolution model. For healthy cell types with homogeneous behavior, 1-2 clusters is adequate; for tumor cells, which can be highly heterogeneous, a higher G is recommended. We use $G=10$ in the tumor analyses in this manuscript.”

Thanks for your response. The authors only provided descriptive guidance for the choice of G rather than giving a concrete table or plot to delineate the impact of G on the performance. Please perform more computational analyses to depict the result with different values of G and provide effective guidance for users.

Results:

1. The authors used 700 μm diameter circular region as spots to generate sequencing and regarded as a known control sample to evaluate four algorithms' performance. However, in the real spatial transcriptomics data, the spot was usually not that large. For example, slide-seq's resolution is 10 μm , and 10x Visium is 50 μm . Therefore, the author should provide additional results to validate the heterogeneity difference between 700 μm and 50 μm . Furthermore, the

author used 700 um to mimic spot level data, and the software might not have sufficient evidence to support the generalizability of SpatialDecon.

This raises a key point unaddressed in the original submission: how is spatialdecon expected to perform in small regions like those of Visium experiments? To address this question, we ran immune deconvolution in a colorectal cancer, profiling regions ranging from 1119 to 145633 square microns, including 15 regions smaller than the 2375 square microns of a 55-um diameter spot. The estimated immune composition in Visium-sized regions was effectively identical to that in larger regions. Three of the smallest regions (1100um² – 170um²) returned slightly different proportions of the same cell types found in larger regions. We have added this demonstration to our supplementary material.

Thank you for the additional data for 50um spot decomposition. Supplementary Fig.3 provides evidence that SpatialDecon can be used for low-resolution deconvolution in terms of oncology data. However, I still suggest the author should at least try Visium data for testing generalizability. Otherwise, the author should claim SpatialDecon was specifically designed for the GeoMx DSP platform and should specify the application scope for the GeoMx DSP data in the title and abstract.

4. For the result of using tumor microenvironment, it is good to identify different immune cell populations (myeloid cell, B cell, CD4 T cell, and CD8 T cell). However, in the real tumor microenvironment, there should be more specific cell populations. For example, CD8 T cells have the subset of effector, exhausted progenitor CD8 T cell, exhausted CD8 T cell. Therefore, the algorithm did not show a significant power in revealing the high-resolution for specific immune cell populations.

This is an important point neglected in our first version. Although the submitted manuscript only showed somewhat coarsely-defined cell types, this limitation results from the safeTME's reliance on Monaco (2019), not on an intrinsic limitation of the algorithm. In other studies, we have found that spatialdecon still works when given a very fine-grained cell profile matrix with >40 cell types.

To demonstrate this point, we assembled a matrix of 44 cell types by combining our fibroblast and endothelial cell profiles with 42 immune cell subtypes reported from a tumor scRNA-seq study (Zilionis 2019). As an example of its granularity, this matrix contains 7 T-cell subtypes and 9 macrophage subtypes.

Although we know of no data that can validate results for any single cell subcluster, we used our benchmarking study to confirm the basic sanity of the algorithm when using granular cell profiles. After performing deconvolution with the 44 cell type matrix, we added the scores from related cell types for comparison to protein. E.g., we compared the total abundance of the 9 macrophage cell types to CD68 protein expression. SpatialDecon's concordance with marker proteins was similarly accurate as we reported with the SafeTME matrix. We now show these results in the supplement and refer to them in the results.

We also applied this more granular cell profile matrix to the tumor shown in Figures 5 & 6. Cell subsets showed distinct and internally consistent spatial distributions. For example, the cycling T-cell cluster, hT7, was primarily present in the B-cell enriched regions, which are likely tertiary lymphoid structures. In contrast, cluster hT5 was almost exclusively found in the upper-right corner of the tumor, away from the B-cells. We now show these results in the supplement and allude to them in the results.

Thanks for the declaration. I agree with you that it is hard to find experimental data to validate the predicted result. However, the author should provide solid references that can validate the results in terms of Fig. 5 and 6.

Codes: 1. The reviewer understands that using formulated mocking data is to save running time. However, it is worthy of putting a real large dataset as an example.

Good idea. The SpatialDecon R library now includes a vignette showing the use of SpatialDecon on the NSLCL tumor described in our manuscript. This update will appear after BioConductor approves our changes. Until then, it can be seen at: https://github.com/Nanostring-Biostats/SpatialDecon/blob/branches/for-journal/vignettes/SpatialDecon_vignette_NSCLC.Rmd. This vignette will generate if the package is installed from the branch "revisions-for-journal".

I successfully install it by using "master" branches. However, I cannot install the package by using "revisions-for-journal." Here is the error message (duplicated vignette title)

```
> devtools::install_github("Nanostring-Biostats/SpatialDecon",
+                           ref = "revisions-for-journal",
+                           build_vignettes = TRUE)
Downloading GitHub repo Nanostring-Biostats/SpatialDecon@revisions-for-journal
√ checking for file 'C:\Users\cyz\AppData\Local\Temp\RTmpI17Fkw/remotes69c954f79e6\Nanostring-Biostats-SpatialDecon-8dbd721\DESCRIPTION' (676ms)
- preparing 'SpatialDecon': (452ms)
√ checking DESCRIPTION meta-information ...
- installing the package to build vignettes
E creating vignettes (1m 50.9s)
  duplicated vignette title:
  'SpatialDecon_vignette'

--- re-building 'SpatialDecon_vignette.Rmd' using rmarkdown
Warning: The vignette title specified in \VignetteIndexEntry{} is different from the title in the YAML metadata. The former is "SpatialDecon_vignette", and the latter is "Use of
the SpatialDecon package in estimating and exploring mixed cell abundance in spatially-resolved gene expression data". If that is intentional, you may set options(rmarkdown.html_v
ignette.check_title = FALSE) to suppress this check.
--- finished re-building 'SpatialDecon_vignette.Rmd'

--- re-building 'SpatialDecon_vignette_NSCLC.Rmd' using rmarkdown
Warning: The vignette title specified in \VignetteIndexEntry{} is different from the title in the YAML metadata. The former is "SpatialDecon_vignette", and the latter is "Use of
the SpatialDecon package in estimating and exploring mixed cell abundance in spatially-resolved gene expression data". If that is intentional, you may set options(rmarkdown.html_v
ignette.check_title = FALSE) to suppress this check.
--- finished re-building 'SpatialDecon_vignette_NSCLC.Rmd'

Error: Duplicate vignette titles.
Ensure that the %\VignetteIndexEntry lines in the vignette sources
correspond to the vignette titles.
Execution halted
Error: Failed to install 'SpatialDecon' from GitHub:
System command 'Rcmd.exe' failed, exit status: 1, stdout + stderr (last 10 lines):
Ex --- finished re-building 'SpatialDecon_vignette.Rmd'
Ex
Ex --- re-building 'SpatialDecon_vignette_NSCLC.Rmd' using rmarkdown
Ex Warning: The vignette title specified in \VignetteIndexEntry{} is different from the title in the YAML metadata. The former is "SpatialDecon_vignette", and the latter is "Use of
the SpatialDecon package in estimating and exploring mixed cell abundance in spatially-resolved gene expression data". If that is intentional, you may set options(rmarkdown.html_v
ignette.check_title = FALSE) to suppress this check.
Ex --- finished re-building 'SpatialDecon_vignette_NSCLC.Rmd'
Ex
Ex Error: Duplicate vignette titles.
Ex Ensure that the %\VignetteIndexEntry lines in the vignette sources
Ex correspond to the vignette titles.
Ex Execution halted
> |
```

2. If a user wants to upload 10X Visium spatial data, is there any portal to load in? If SpatialDecon cannot receive other data types, the generalizability might be limited.

The package supports Visium and GeoMx with equal convenience. For either platform, the user need only extract 1. An expression matrix, and 2. The estimated background level in each sampled region. (In Visium, as discussed in more detail after a question from Reviewer 2, expected background is just 0.)

Thanks for your clarification. For the usability, I still suggest 1. Add wrapped function for directly inputting the output from Visium platform; 2. adding one sample on vignettes to show how SpatialDecon input from Visium data.

3. The visualization function (for example, the sector distribution chart) might be packaged and form an individual function so that it can be easily used.

We appreciate this feedback, and we are already working on a larger visualization package that includes the function alluded to above.

Due to I cannot download and use the package from the "revisions-for-journal" branch, so I cannot see the changes on the well-packaged function.

4. It will be interesting to have a connection between the proposed tool with the existing analysis tools. For example, if there is a functional can extract the cell abundance score and add it to the Seurat object as a metafile, it will be very helpful.

Good idea. We have added an S4 method to our package that facilitate applying spatialdecon to Seurat S4 objects. This update will appear after BioConductor approves our changes. Until then, you can see the new code at <https://github.com/Nanostring-Biostats/SpatialDecon/blob/revisions-for-journal/R/runspatialdecon.R>.

Thanks for the changes. The "revisions-for-journal" branch is still not available for me.

We would like to thank both reviewers for their continued engagement and useful suggestions. This resubmission addresses every comment made by the reviewers. The result is numerous edits to the main text, new features and vignettes in the SpatialDecon R library, and 6 new supplemental figures. We believe these updates have strengthened the manuscript considerably, and we are grateful for the thorough reviews that led to them.

Below, please find reviewer comments in black and our responses in blue.

Reviewer 1:

Methods 1. The authors only used one dataset to demonstrate the rationale for log-transformation, which is not sufficient to draw the conclusion. Please provide more data to demonstrate the claim. In addition, the distribution of noise in the data is to some extent impacted by the sequencing techniques.

Good idea. We have created a new supplemental figure, Figure S2, which scrutinizes the argument for log-transforming GeoMx data. As in TCGA, we find that linear scale data has high skewness and extreme heteroscedasticity, and that log-transformation greatly mitigates both these conditions. The main text refers to these results.

We agree that different technologies will have different distributions of technical noise. However, since biological variability dominates technical noise (genomics platforms would not be very useful if this were not the case), we argue that each platform's distinct probability distribution of technical noise will not substantially impact any deconvolution method.

The deconvolution methodology in reference to Zhong & Liu (2012) was dedicated to microarray data rather than RNA-Seq or scRNA-Seq, which may not be suitable to describe the newly-developed sequencing techniques. Therefore, it is necessary to discuss whether the linear mean model is still suitable for the current sequencing data. Please clarify the rationale comprehensively via theoretical proof or references.

We have added a supplementary notes section entitled "Rationale for the linear mean model used by SpatialDecon". This section reviews the argument of Zhong & Liu (2012) and it shows why the theory from their paper still applies to spatial data. The main text refers to this discussion.

Methods 3. After thresholding Y using epsilon, all the values smaller than epsilon obtain the same value, i.e., epsilon. This operation will remove variations within the data which may result from both biological or technical factors. Please perform more computational analyses to demonstrate that different values of epsilon will not drastically affect the result of Algorithm1.

We have run a new experiment evaluating the impact of epsilon on Algorithm 1. Its results are shown in the new Supplemental Figure 7. We find that Algorithm 1's accuracy changes not at all for a wide range of values around the default epsilon. Accuracy drops dramatically for extremely high values of epsilon, but these values are far beyond the default. The main text refers to these results.

Methods 4.

4. In the second step of Algorithm 1, the beta matrix was optimized in a column-wise manner separately. This relies on a hypothesis that the columns are independent with each other. However, the authors have not made such a hypothesis in this manuscript, nor do they provide relevant references to explain the underlying rationale.

Good point: we see how this could cause reader confusion. We now state in the text that "Separate optimization over columns of Y is permissible since different columns of Y represent data from different region of a tissue, and so can be treated as effectively independent."

Thanks for your response. However, the authors only revised this step in Algorithm 1 by adding a descriptive sentence rather than providing solid computational or reference evidence. As known, even different regions of tissue may impact each other via signaling pathways, e.g., microglia and astrocytes in the spinal cord. Hence, this statement is qualified as a rationale for step 2 in Algorithm 1.

Good point. We agree that cell types will change expression patterns based on a variety of factors, including tissue region. Our assumption of independence, then, is a necessary simplification of the far more complex biological reality. We have now amended the manuscript to recognize this caveat, writing, "This step assumes that different columns of Y, i.e. different tissue regions, are statistically independent. To the extent that this assumption fails, greater accuracy could be gained by modelling the dependence between regions. Modelling this dependence structure is a problem left for future algorithms; to our knowledge, no attempts to do so have been described." We speculate that in practice, modelling dependence between regions would lead to improved statistical efficiency but not to dramatic differences in algorithm performance.

Methods 5. Please cite relevant references regarding standard normal distribution or theoretical proof as a support of this step.

We have added a reference proving the asymptotic normality of the test statistic in question: Harrell, Frank E., Jr. (2001). "Section 9.3.1". Regression modeling strategies. New York: Springer-Verlag. ISBN 0387952322.

Methods 6. Since the results are nearly identical between with and without outlier removal, I am not convinced of the performance of outlier removal in SpatialDecon. Please conduct experiments to demonstrate that the outlier removal step does decrease the outliers in data. Otherwise, this step is unnecessary to the performance of SpatialDecon. For the latter situation, please remove this step.

Good idea. We have added a new supplemental figure (Supplementary Figure 8) examining the behavior of our outlier removal operations. To summarize, we re-ran the marker protein validation experiment, adding noise to the data from a subset of 30 genes. We found 1. the perturbed genes were flagged as outliers at a much-increased rate, and 2. removing outliers improved deconvolution performance by a

modest but statistically significant amount. With this confirmation that outlier removal works as intended and improves performance, we propose to retain this step in our algorithm. The main text refers to these results.

Methods 7 iii. (re outlier removal). The authors did not describe the rationale for this operation in the response of neither question 6 nor question 7.

We apologize for misinterpreting this comment in the last submission. We now understand the question to pertain to how we chose a threshold of 3 for outlier removal.

This threshold is measured on the scale of log₂ fold-changes between fitted and observed expression levels. We cannot claim that the default value of 3 is optimal, but it is reasonable: a cutoff of 3 corresponds to an observed value 8-fold away from the expected value, which is a substantial change in gene expression.

Most importantly, our default value of 3 performs well, and the algorithm is not highly sensitive to the choice of this threshold. To demonstrate the impact of the outlier removal threshold, we ran our benchmarking study using a range of values of this threshold (Supplementary Figure 8b). We found that average correlation between SpatialDecon and canonical marker protein expression varied little with the choice of threshold, varying from a correlation of 0.70 achieved with a threshold of 2 to a correlation of 0.66 achieved with a threshold of 6. Our default threshold of 3 achieved a correlation of 0.69 and was not statistically significantly worse than the best-performing threshold of 2 (paired t-test $p = 0.23$). The main text refers to these results.

Methods 7 iv. The authors only provided descriptive guidance for the choice of G rather than giving a concrete table or plot to delineate the impact of G on the performance. Please perform more computational analyses to depict the result with different values of G and provide effective guidance for users.

Good idea. The new Supplementary Figure 9 explores the impact of G on deconvolution performance. To summarize, we ran SpatialDecon on each tissue in our benchmarking data using a range of values of G. We found G to have negligible impact on deconvolution performance. The main text refers to these results.

In light of these results, we conclude that our original guidance is appropriate: “For healthy cell types with homogeneous behavior, 1-2 clusters is adequate; for tumor cells, which can be highly heterogeneous, a higher G is recommended.” This guidance achieves good performance in our benchmarking dataset, and it will guard against future datasets with substantial intra-tumor heterogeneity.

Results 1. Thank you for the additional data for 50um spot decomposition. Supplementary Fig.3 provides evidence that SpatialDecon can be used for low-resolution deconvolution in terms of oncology data.

However, I still suggest the author should at least try Visium data for testing generalizability. Otherwise, the author should claim SpatialDecon was specifically designed for the GeoMx DSP platform and should specify the application scope for the GeoMx DSP data in the title and abstract.

This is a good point. The supplemental material now includes an application to Visium and an application to Spatial Transcriptomics (the pre-commercial version of Visium).

The new Supplementary Fig. 11 shows SpatialDecon results from an ovarian cancer Visium dataset. The dominant immune cell phenotypes are consistent with the other tumors profiled in this study, but without tertiary lymphoid structures.

The new Supplementary Fig. 10 shows SpatialDecon results from a breast cancer Spatial Transcriptomics dataset. Two findings from this analysis support SpatialDecon's use in Visium data. First, the overall abundance and distribution pattern of immune cells resembles the coherent biological picture of the NSCLC sample from our Figures 5&6: B-cells are confined to a few dense pockets, T-cells are abundant near B-cells but invade in low levels across the tissue, and macrophages are spread widely but at low abundance across the tissue. Second, the regions with the greatest B-cell and T-cell abundance scores were classified by a pathologist in the original study as typified by "inflammatory cells." The main text refers to these results.

Results 4. I agree with you that it is hard to find experimental data to validate the predicted result. However, the author should provide solid references that can validate the results in terms of Fig. 5 and 6.

Good idea. We have found references demonstrating that several of the T-cell sub-clusters we observe behave consistent with known biology. We have added the below text to the manuscript:

"Sub-clusters' spatial distributions were distinct and consistent with known biology. For example, the cycling T-cell cluster, hT7, was primarily present in the B-cell enriched regions. The dense T-cells and B-cells of these regions suggest they are tertiary lymphoid structures, where T-cells are activated and prompted to proliferate (Goc 2014). Another T-cell cluster confined to these regions was, hT4, which is defined by expression of CXCL13, a B-cell chemoattractant pivotal in forming and maintaining tertiary lymphoid structures (Pimenta 2014). Finally, the cytotoxic CD8 T-cell cluster, hT1, and the Treg cluster, hT3, invaded the same regions of the tissue, consistent with their demonstrated tendency to traffic together (Feng 2017)."

Codes 1. I successfully install it by using "master" branches. However, I cannot install the package by using "revisions-for-journal." Here is the error message (duplicated vignette title).

We have fixed this error and confirmed that the package installs with the below call:

```
devtools::install_github("Nanostring-Biostats/SpatialDecon",  
  ref = "revisions-for-journal",  
  build_vignettes = FALSE)
```

Codes 2. For the usability, I still suggest 1. Add wrapped function for directly inputting the output from Visium platform; 2. adding one sample on vignettes to show how SpatialDecon input from Visium data.

Good idea. We have created a S4 method for applying SpatialDecon to a Seurat visium object, and we have added a new vignette applying SpatialDecon to a Visium dataset.

Codes 3. Due to I cannot download and use the package from the “revisions-for-journal” branch, so I cannot see the changes on the well-packaged function.

We hope our latest updates will allow for successful installation. You can find our visualization functions in the package’s “R” folder, under TIL_barplot.R and florets.R. The package vignette shows the use of both functions.

Codes 4. Thanks for the changes. The “revisions-for-journal” branch is still not available for me.

This branch should now be public.

Reviewer 2:

The authors have answered all my comments and included SpatialDWLS (RCTD and SPOTlight were not included for valid reasons). While I am satisfied with many of their answers, there are still a few key aspects that, in my opinion, need improvement.

COMMENTS TO THE MANUSCRIPT

Their statement “We named cell clusters using published marker genes, curated to better reflect the known biology of each cell type based on the scientific literature” is still too vague (references?).

Furthermore, even though they provide cell profile matrices and the marker genes used to classify clusters (Supp. Table 4), a clear methodology that can be reproduced is needed (“In each dataset, we derived cell clusters”... how?).

We have updated this section of our manuscript in a way such that the above comments no longer apply. Specifically, we have substantially revised and expanded our library of cell profile matrices. We opted to use only datasets where all cell types had been annotated by their creators. The Methods and Results sections have been updated to reflect this new approach. The above comments apply to derivation and naming of cell clusters, which we no longer undertake.

Their section “Details on methods from Danaher & Kim (2020)” (<https://github.com/Nanostring-Biostats/CellProfileLibrary>) should be introduced in the manuscript (Methods section).

Our apologies, but we do not understand this request. The section “Details on methods from Danaher & Kim (2020)” in the github readme file is copied verbatim from the manuscript’s Methods section, as this was our most complete description of the cell profile matrices’ derivation. Should we refer to this section of the readme in the manuscript? For now, we have opted to clarify the source of this material in the github readme file, which now identifies that material as a quote from the manuscript.

The scaling to the same 80th percentile, 90th or median is still not clear quantitatively (e.g. “the 90th percentile fell within the robustly expressed genes but left enough genes above it”). How many are “enough genes”? For instance, if authors require ≥ 10 genes as a threshold, this could indeed be a different percentile per cell type. This insight needs to be clarified to further investigate new datasets with their method.

We understand this line of questioning to have originally encompassed two topics. First, there was a question about how we chose between arithmetic means, geometric means and medians to collapse multiple replicates into a single representative expression profile. We believe the first concern has been resolved by the new Methods subsection, “Capturing central tendencies”.

The second concern, still salient in this round of reviews, concerns the choice of 80th and 90th percentiles for different purposes. We discuss each in turn below:

1. In our Algorithm 3, we scale the expression profiles of pure tumor regions so their 90th percentile matches the 90th percentile of the of the cell types already in the cell profile matrix. This rescaling need not be accurate: its role is to avoid convergence failures by putting the columns of the cell profile matrix on roughly the same scale. We have updated the manuscript to explain the role of this rescaling and to caution against over-interpretation of the scale of the tumor profiles' deconvolution results. It now states, "To avoid convergence failures, scale this profile similarly to the other columns of the cell profile matrix such that its 90th percentile is equal to the average 90th percentile of the other columns." And, "The scaling operation in step 3 is arbitrary. Therefore, it is not recommended to directly compare tumor abundance scores derived with this method to immune abundance scores. This algorithm should primarily be considered a tool for obtaining more accurate immune abundance scores."
2. To normalize the flow-sorted RNAseq data of Gentles et al., we used the 90th percentile. We have updated the manuscript to further clarify the parameters guiding this choice: "In this dataset, the 90th percentile fell within the robustly expressed genes, exceeding 2 TPM in every sample, but left over a thousand genes above it to avoid excessive influence by noise in a small number of extreme high expressers. For the purposes of deriving an average profile, any standard RNAseq normalization method would have sufficed."
3. In the derivation of the SafeTME matrix, we scale similar cell types' expression profiles by their 80th percentiles to place results from different datasets on the same scale. We found no solution but to make this choice arbitrarily from within a reasonable range of values. The manuscript now expands on the impact of this choice: "This choice of a scaling factor is necessarily arbitrary, and error in its selection causes bias when comparing the abundance of one cell type to another. If we had chosen 0.7, the SafeTME would return mast cell abundances 40% higher and macrophage abundances 120% higher. If we had chosen 0.9, we would see mast cells 43% lower and macrophages 23% lower. Due to the uncertainty in scaling different cell type's profiles, comparison of one cell type's abundance to another's is always fraught; the need to compute mast cell and macrophage scaling factors across datasets makes their relative abundance estimates subject to additional error."

To summarize: this choice of scaling factor adds uncertainty around the estimated abundance of mast cells and macrophages relative to other cell types in the SafeTME. Fortunately, we have observed both these cell types to return abundances consistent with expected biology, suggesting our choice of scaling factor was not far from the correct value.

Finally, during their rebuttal, the authors removed the words "in spatial gene expression data" from their section "Log-normal regression improves deconvolution performance". Since this is no longer exclusive of spatial transcriptomics, it would be great for the field if the authors analyzed and comment novelty, similarities and differences with dtangle (Hunt et al., Bioinformatics 2019). Specifically, with respect to this paragraph from Hunt et al.: "dtangle's approach is built on a biologically appropriate linear mixing model of linear-scale expressions but robustly fitting the model using log-transformed data and thus sets it apart from other deconvolution methods."

Good idea. Dtangle shares SpatialDecon's linear mean model and log-scale variance model. A crucial difference is that dtangle is a marker-gene-based approach, while SpatialDecon introduces this data

generating model to the family of regression-based deconvolution methods. We have added the below text to the manuscript.

“Combining a linear-scale mean model and log-scale variability was proposed before as part of the dtangle algorithm, which depends on cell type-specific marker genes. SpatialDecon assumes the same data generating model as dtangle, but its regression framework harnesses all genes, regardless of cell type specificity.”

COMMENTS TO THE CODE

Their main command (from Github) to install the package does not work (I tried in 2 laptops, both mac OS and Linux):

```
## COMMAND
devtools::install_github("Nanostring-Biostats/SpatialDecon", ref = "master", build_vignettes = TRUE)
```

```
## LOG
--- re-building 'SpatialDecon_vignette.Rmd' using rmarkdown
Error: processing vignette 'SpatialDecon_vignette.Rmd' failed with diagnostics:
object 'is_R_CMD_check' not found
--- failed re-building 'SpatialDecon_vignette.Rmd'
```

SUMMARY: processing the following file failed:
'SpatialDecon_vignette.Rmd'

```
Error: Vignette re-building failed.
Execution halted
Error: Failed to install 'SpatialDecon' from GitHub:
System command 'R' failed, exit status: 1, stdout + stderr (last 10 lines):
E> --- re-building 'SpatialDecon_vignette.Rmd' using rmarkdown
E> Error: processing vignette 'SpatialDecon_vignette.Rmd' failed with diagnostics:
E> object 'is_R_CMD_check' not found
E> --- failed re-building 'SpatialDecon_vignette.Rmd'
E>
E> SUMMARY: processing the following file failed:
E> 'SpatialDecon_vignette.Rmd'
E>
E> Error: Vignette re-building failed.
E> Execution halted
```

Our reading suggests this may be an issue with rmarkdown versions; namely, rmarkdown versions < 2.6 will produce this error message.

To guard against this error, which we imagine will impact many users, we have updated our installation instructions to omit vignette building. We have confirmed that the below command works:

```
devtools::install_github("Nanostring-Biostats/SpatialDecon",  
  ref = "revisions-for-journal",  
  build_vignettes = FALSE)
```

Furthermore, on the “Code and Software Checklist PDF (154KB)” I saw the package was already in Bioconductor with requirements $R \geq 4.0.0$. However, I tried that way too and it didn’t work:

```
## COMMAND
```

```
BiocManager::install("SpatialDecon")
```

```
## LOG
```

```
Bioconductor version 3.11 (BiocManager 1.30.16), R 4.0.1 (2020-06-06)
```

```
Installing package(s) 'SpatialDecon'
```

```
Warning message:
```

```
In .inet_warning(msg) :
```

```
package ‘SpatialDecon’ is not available (for R version 4.0.1)
```

We have corrected this error: the package now requires 4.1.

Nevertheless, I was able to load the different functions and data objects to test their vignette from Bioconductor

(https://bioconductor.org/packages/release/bioc/vignettes/SpatialDecon/inst/doc/SpatialDecon_vignette.R) and, fortunately, that did work.

Thank you for the extra effort.

REVIEWER COMMENTS

Reviewer #1 (Remarks to the Author):

Thanks for your responses to my previous comments and the reviewer can clearly see your efforts. We still have some concerns about only using one dataset to make a conclusion. Please check our specific comments in the attached file (highlighted in orange).

Reviewer #2 (Remarks to the Author):

The authors have carefully addressed all my previous remarks and the package can be successfully installed from their Github repository. The manuscript is, in my opinion, suitable for publication in its current form.

Methods 1. The authors only used one dataset to demonstrate the rationale for log-transformation, which is not sufficient to draw the conclusion. Please provide more data to demonstrate the claim. In addition, the distribution of noise in the data is to some extent impacted by the sequencing techniques.

Good idea. We have created a new supplemental figure, Figure S2, which scrutinizes the argument for log-transforming GeoMx data. As in TCGA, we find that linear scale data has high skewness and extreme heteroscedasticity, and that log-transformation greatly mitigates both these conditions. The main text refers to these results.

Thanks for your response. However, the authors have not fixed this issue directly since they did not explain why they only used the microenvironment regions of the NSCLC tumor to demonstrate the rationale for log-transformation. Other datasets should be added to support the same conclusion here.

We agree that different technologies will have different distributions of technical noise. However, since biological variability dominates technical noise (genomics platforms would not be very useful if this were not the case), we argue that each platform's distinct probability distribution of technical noise will not substantially impact any deconvolution method.

As to the impacts of sequencing technologies, please cite relevant references to back up the conclusion.

The deconvolution methodology in reference to Zhong & Liu (2012) was dedicated to microarray data rather than RNA-Seq or scRNA-Seq, which may not be suitable to describe the newly-developed sequencing techniques. Therefore, it is necessary to discuss whether the linear mean model is still suitable for the current sequencing data. Please clarify the rationale comprehensively via theoretical proof or references.

We have added a supplementary notes section entitled "Rationale for the linear mean model used by SpatialDecon". This section reviews the argument of Zhong & Liu (2012) and it shows why the theory from their paper still applies to spatial data. The main text refers to this discussion.

Thanks for your response. I am fine with the explanation of using the linear mean model to formulate transcript counts. However, it still demands more description about the suitability of log-transformation for noise.

Methods 6. Since the results are nearly identical between with and without outlier removal, I am not convinced of the performance of outlier removal in SpatialDecon. Please conduct experiments to demonstrate that the outlier removal step does decrease the outliers in data. Otherwise, this step is unnecessary to the performance of SpatialDecon. For the latter situation, please remove this step.

Good idea. We have added a new supplemental figure (Supplementary Figure 8) examining the behavior of our outlier removal operations. To summarize, we re-ran the marker protein validation experiment, adding noise to the data from a subset of 30 genes. We found 1. the perturbed genes were flagged as outliers at a much-increased rate, and 2. removing outliers improved deconvolution performance by a modest but statistically significant amount. With this confirmation that outlier removal works as

intended and improves performance, we propose to retain this step in our algorithm. The main text refers to these results.

Thanks for your response. Now that only one benchmarking dataset was used for comparison, it is not enough to highlight the advantage of SpatialDecon over the method without outlier removal. The reviewer suggests repeating the same comparison on two new datasets to make sure of scientific rigor.

Results 1. Thank you for the additional data for 50um spot decomposition. Supplementary Fig.3 provides evidence that SpatialDecon can be used for low-resolution deconvolution in terms of oncology data. However, I still suggest the author should at least try Visium data for testing generalizability. Otherwise, the author should claim SpatialDecon was specifically designed for the GeoMx DSP platform and should specify the application scope for the GeoMx DSP data in the title and abstract.

This is a good point. The supplemental material now includes an application to Visium and an application to Spatial Transcriptomics (the pre-commercial version of Visium).

The new Supplementary Fig. 11 shows SpatialDecon results from an ovarian cancer Visium dataset. The dominant immune cell phenotypes are consistent with the other tumors profiled in this study, but without tertiary lymphoid structures.

The new Supplementary Fig. 10 shows SpatialDecon results from a breast cancer Spatial Transcriptomics dataset. Two findings from this analysis support SpatialDecon's use in Visium data. First, the overall abundance and distribution pattern of immune cells resembles the coherent biological picture of the NSCLC sample from our Figures 5&6: B-cells are confined to a few dense pockets, T-cells are abundant near B-cells but invade in low levels across the tissue, and macrophages are spread widely but at low abundance across the tissue. Second, the regions with the greatest B-cell and T-cell abundance scores were classified by a pathologist in the original study as typified by "inflammatory cells." The main text refers to these results.

Thanks for the update. I can tell at least SpatialDecon can work with Visium and ST dataset, which may be an interesting find and showing a potential application in a broad field. However, I did not find where the author cited supplementary figure 11 in the manuscript. Please cite the supplementary file and add appropriate supportive references for demonstrating decomposition results.

Codes 2. For the usability, I still suggest 1. Add wrapped function for directly inputting the output from Visium platform; 2. adding one sample on vignettes to show how SpatialDecon input from Visium data.

Good idea. We have created a S4 method for applying SpatialDecon to a Seurat visium object, and we have added a new vignette applying SpatialDecon to a Visium dataset.

Thanks for your response. I see the change in the vignette. However, there is a minor misuse of the Seurat package for importing Visium data in your vignette named "SpatialDecon_vignette_ST". The author should use Load10X_spatial to import Visium data instead of "CreateSeuratObject."

To the editors and our reviewer:

We would like to thank the reviewer for their continued time and effort. These cumulative reviews have resulted in a more complete and convincing manuscript, and we are grateful for them. We have addressed each of these latest comments. In doing so, we made some changes to the main text, and we made extensive additions to the supplement, including a new section of text and 2 new figures, both of which include results from 2 new datasets.

Below, please find reviewer comments in orange and our responses in black.

Thank you, on behalf of all the authors,

Patrick Danaher, Ph.D.

Thanks for your response. However, the authors have not fixed this issue directly since they did not explain why they only used the microenvironment regions of the NSCLC tumor to demonstrate the rationale for log-transformation. Other datasets should be added to support the same conclusion here.

Regarding further datasets:

Good point. We have repeated our analysis of skewness and heteroscedasticity in two new datasets, one from healthy pancreas and one from healthy kidney, both using the GeoMx Whole Transcriptome Atlas. These analyses recapitulate our previous results: on the linear scale, high-expression genes have much higher variance than low-expression genes, and all genes are highly right-skewed. Log-transformation either corrects or greatly reduces the scope of these problems. We have added these results to our Supplementary Material, and we have excerpted them at the end of this response.

Regarding our use of only microenvironment regions:

Good point; we should have further explained our choices. We used the microenvironment regions because we worried the profound differences between tumor and microenvironment gene expression would increase variance and make linear-scale results look worse than they should. To convey that this choice was not arbitrary, the supplementary material now explains, "Tumor regions were excluded due to concerns that the profound differences between tumor and microenvironment regions would cloud interpretation of results."

The figure below shows a quick comparison of how this analysis would look using all regions instead of using only microenvironment regions. Our key observation from the original analysis was that linear-scale data has greater heteroscedasticity; this observation strengthens when tumor and microenvironment regions are analyzed jointly. The range of SDs is from 0-1558 when all regions are considered, and from 0-997 when only the microenvironment is considered.

Another key observation in our original analysis was that linear-scale data was positively skewed. Comparative results from this analysis are below. In the complete dataset, the average gene had skewness of 1.09 on the linear scale; in the microenvironment-only dataset, this was only 0.92.

As to the impacts of sequencing technologies, please cite relevant references to back up the conclusion.

This comment relates to our claim that biological variability tends to overwhelm variability from technical noise. The below references support this claim:

1. A reproducibility study of Spatial Transcriptomics reported a correlation of 0.97 between technical replicates. (Stark, R., Grzelak, M., & Hadfield, J. (2019). RNA sequencing: the teenage years. *Nature Reviews Genetics*, 20(11), 631-656.)
2. A reproducibility study of GeoMx reported that when comparing GeoMx profiling vs. bulk nCounter profiling of cell lines, “Most targets (24 of 27) had R^2 values above 0.75, with a range of correlation values from 0.76 to 0.97.” (Merritt, C. R., Ong, G. T., Church, S. E., Barker, K., Danaher, P., Geiss, G., ... & Beechem, J. M. (2020). Multiplex digital spatial profiling of proteins and RNA in fixed tissue. *Nature biotechnology*, 38(5), 586-599.)

In addition, the supplement now cites these papers and summarizes our discussion of platform effects.

Thanks for your response. I am fine with the explanation of using the linear mean model to formulate transcript counts. However, it still demands more description about the suitability of log-transformation for noise.

We have taken this opportunity to clarify and expand on the manuscript's case for log-scale variance model in a new section in the supplement, entitled "Rationale for the log-scale variance model used by SpatialDecon".

To summarize our argument:

- Least squares methods implicitly assume genes are unskewed and have comparable variance.
- We find in 4 separate datasets that linear-scale gene expression has highly unequal variance and a strong tendency to positive skewness. We find that log-scale gene expression largely corrects these undesirable behaviors.
- We therefore conclude that a log-scale variance model will perform better, mainly because it will avoid overweighting high expressing/highly variable genes, and also because it will correctly weight positive and negative residuals.
- These conclusions play out in Figure 2 in the main text, which shows that 1. Linear-scale variance models perform poorly, and 2. This poor performance can be attributed to a small number of genes with extremely high influence on model results.

In further support of the above argument, DWLS, a linear-scale variance model that adaptively weights genes to account for unequal variance, does not assign excessive influence to any genes in Figure 2. Log-normal regression and DWLS's approach can be considered alternative approaches to addressing unequal variance. Log-normal regression slightly outperforms DWLS in Figure 2, and SpatialDecon substantially outperforms SpatialDWLS in Figure 5. We nonetheless imagine that a future DWLS-like approach could produce satisfactory results in spatial gene expression data.

We would also like to clarify that we do not rely on a claim that gene expression is precisely a log-normal phenomenon. SpatialDecon is motivated by the much more narrow claim that gene expression's unequal variance and skewness make a log-scale variance model more statistically efficient than a linear-scale model.

Thanks for your response. Now that only one benchmarking dataset was used for comparison, it is not enough to highlight the advantage of SpatialDecon over the method without outlier removal. The reviewer suggests repeating the same comparison on two new datasets to make sure of scientific rigor.

To further interrogate the performance of outlier removal in SpatialDecon, we used the GeoMx whole transcriptome kidney and pancreas studies mentioned earlier. From each of these datasets, we simulated error across 1000 genes, yielding a “perturbed” dataset. We then applied SpatialDecon to both the original and the perturbed datasets, first retaining outliers and then removing outliers. “Outlier-retained” results were compared across the original and perturbed datasets, as were “outlier-removed” results. (New Supplementary Figure 10.)

We found that the perturbed genes were flagged at higher rates than the unperturbed genes, confirming the accuracy of our outlier detection procedure. We also found that removing outliers makes the results from perturbed data better resemble the results from unperturbed data, as measured both by correlation and MSE.

These analyses in two new datasets confirm our earlier observations that outlier removal improves accuracy in the presence of outliers. Therefore we propose to retain this feature in the code.

Thanks for the update. I can tell at least SpatialDecon can work with Visium and ST dataset, which may be an interesting find and showing a potential application in a broad field. However, I did not find where the author cited supplementary figure 11 in the manuscript. Please cite the supplementary file and add appropriate supportive references for demonstrating decomposition results.

Good point. We now refer to this figure in the manuscript as part of our discussion of applications to Visium. In the supplemental commentary on this figure (formerly Supplementary Figure 11, now Supplementary Figure 13), we have added three references supporting the biological plausibility of the immune and stroma cell types we found to be most abundant.

Thanks for your response. I see the change in the vignette. However, there is a minor misuse of the Seurat package for importing Visium data in your vignette named “SpatialDecon_vignette_ST”. The author should use `Load10X_spatial` to import Visium data instead of “`CreateSeuratObject`.”

Our apologies, but we may be unable to make this change. The vignette downloads data from the github site from Andersson et al., “Spatial deconvolution of HER2-positive breast tumors reveals novel intercellular relationships.” They do not supply the h5 files expected by `Load10X_spatial`; instead, they supply .tsv.gz files. Please advise if we are missing a solution or if another best practice can be employed while loading data into our vignette.

REVIEWERS' COMMENTS

Reviewer #1 (Remarks to the Author):

The authors addressed all my comments.

Reviewer comments:

Reviewer #1 (Remarks to the Author):

The authors addressed all my comments.

Author response:

We would again like to express our appreciation for the reviewer's time and effort. The manuscript has strengthened considerably as a result of their efforts.